# The Historical Representation and Near Future (2050) Projections of the Coral Sea Current System in CMIP6 HighResMIP

**Jodie A. Schlaefer[1], Clothilde Langlais[2], Severine Choukroun[3], Mathieu Mongin[2], Mark E. Baird[2]**

[1]Commonwealth Scientific and Industrial Research Organisation (CSIRO), Environment, Townsville, Queensland, Australia

[2]CSIRO, Environment, Hobart, Tasmania, Australia

[3]The Centre of Tropical Water and Aquatic Ecosystem Research (TropWATER), Townsville, Queensland, Australia

*Correspondence to*: Jodie A. Schlaefer (jodie.schlaefer@csiro.au)

**Abstract**

The Coral Sea houses expansive coral reefs. Reef health is inextricably linked to water temperatures, which are regulated by the hydrodynamic environment. The ocean current system in the Coral Sea is dominated by jets of the South Equatorial Current (SEC): the North Vanuatu Jet (NVJ), the North Caledonian Jet (NCJ) and the South Caledonian Jet (SCJ). We investigated the projected near-future (2050) changes in the temperature and transport structure of the Coral Sea using the three highest
resolution climate models from the CMIP6 HighResMIP experiment. We found that the HighResMIP models successfully represented the historical temperature and transport structure of the SEC jets, and their El Niño Southern Oscillation related variability. Surface ocean warming of 0.78°C and 1.12°C was projected in the Coral Sea under 1.5°C and 2°C global air temperature warming, respectively. The maximum depth of the warming signal deepened by 30 m per decade, penetrating to 400 m by 2050. This indicated the additional thermal stress that could be experienced by Coral Sea ecosystems. Interestingly,
the surface warming was associated with a sub-surface cooling between 400 and 600 m. Decreases in the transports of the NVJ and NCJ, and an intensification of the SCJ were also projected in the HighResMIP models. The magnitudes of the changes were relatively small (2% to 7% of historical means), and of a similar order to the variability in transport associated with the El Niño Southern Oscillation. Our analysis further showed that the transport projections of the NVJ and NCJ varied with depth, where surface intensifications coincided with the areas of greatest warming. These changes could modify western boundary
currents and upwelling dynamics on the Great Barrier Reef shelf.

**1 Introduction**

The Coral Sea, in the western tropical Pacific, is framed by Vanuatu on its eastern boundary, by the Solomon Islands and Papua New Guinea on its northern boundary and by the coast of Queensland, Australia on its western boundary. The area
includes the Coral Sea Marine Park and the Great Barrier Reef (GBR) Marine Park, which together house approximately 48,000 km$^2$ of shallow water coral reefs, as well as expansive mesophotic reefs (Bridge et al., 2019). The Coral Sea has a complex system of currents driven by the South Equatorial Current (SEC), which is the westward portion of the South Pacific Gyre (Ganachaud et al., 2014; Figure 1). The SEC is steered by topographic features around Vanuatu and New Caledonia as it enters the Coral Sea, dividing it into the North Vanuatu Jet (NVJ), the North Caledonian Jet (NCJ) and the South Caledonian
Jet (SCJ; Ganachaud et al., 2014; Kessler & Cravatte, 2013). The NVJ, NCJ and SCJ continue westward before bifurcating into western boundary currents along the Australian continental shelf that houses the outer reefs of the GBR.

Under climate change, high uncertainties remain in the projected changes of the SEC transport. Sub-tropical gyres, including the South Pacific Gyre that the SEC forms part of, are expected to shoal and accelerate as warming increases stratification (Peng et al., 2022). However, investigations of future changes to the transport of the SEC close to the equator have reported
both reductions (Ganachaud et al., 2011) and increases (Sen Gupta et al., 2012). The increase was attributed to the projected intensification of the south-easterly trade winds that drive the South Pacific Gyre, rather than temperature effects (Sen Gupta

et al., 2012). Of particular interest for this study, changes in the SEC jets that reach the Coral Sea (i.e., the NVJ, the NCJ and the SCJ) have not been explicitly investigated.

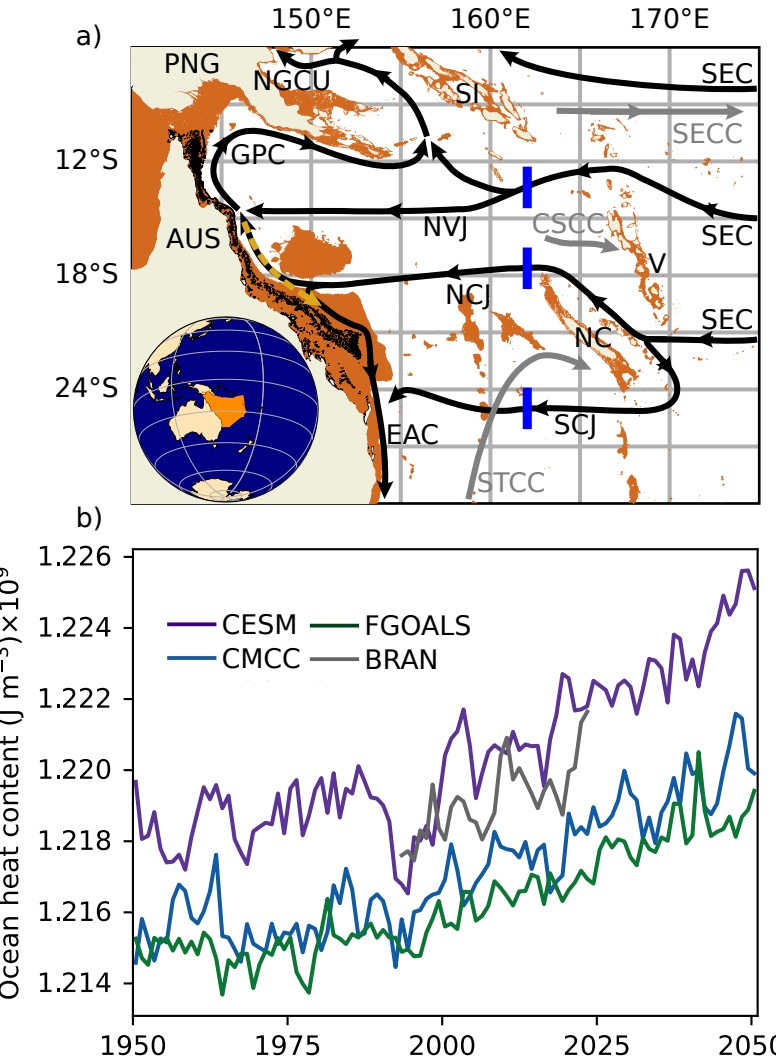

Figure 1. The Coral Sea. a) Map of the currents in the region (top 1000 m) derived from Cravatte et al. (2021) and Ganachaud et al. (2014) including the SEC and its jets, the North Vanuatu Jet (NVJ), North Caledonian Jet (NCJ) and South Caledonian Jet (SCJ) (black arrows). Western boundary currents (black and yellow dashed arrows): New Guinea Coastal Undercurrent (NGCU), Gulf of Papua Current (GPC) and East Australian Current (EAC). Counter currents (grey arrows): South Equatorial Counter Current (SECC), Coral Sea Counter Current (CSCC) and South Pacific Subtropical Counter Current (STCC). Islands: Vanuatu (V), Solomon Islands (SI), New Caledonia (NC), Australia (AUS) and Papua New Guinea (PNG). 0 to 1000 m bathymetry (Orange; from GEBCO 2023 bathymetry [GEBCO Compilation Group, 2024]) and the Great Barrier Reef (black) are shown. The blue lines indicate the three sections visualized in Figure 4. The inset shows the Coral Sea area, as defined by the International Hydrographic Organization (Flanders Marine Institute, 2018). b) Annual average ocean heat content of the top 200 m of the Coral Sea.

The SEC jets in the Coral Sea have strong inter-annual variability associated with the El Niño Southern Oscillation (ENSO; Cravatte et al., 2021), and it is important to consider the magnitude of future trends within the context of this inherent variability (e.g., Sen Gupta et al., 2012). ENSO is associated with a tropical Pacific wind and sea surface temperature pattern that cycles through El Niño, neutral, and La Niña phases (Calvo et al., 2007; Picaut et al., 1996; Risbey et al., 2009). The SEC jets

strengthen a few months after El Niño and weaken a few months after La Niña (Kessler and Cravatte, 2013a).

The health of the reef ecosystems in the Coral Sea and GBR is intrinsically linked to the hydrodynamics. For example, localized upwellings where the SEC jets interact with the continental shelf break can increase nutrient availability, fostering phytoplankton bloom (Furnas and Mitchell, 1996; Wolanski et al., 1988) and can generate relatively cool refugia in conditions where corals would otherwise bleach due to thermal stress (Riegl and Piller, 2003; Spring and Williams, 2023). These refugia

may become increasingly important against the backdrop of more frequent marine heat waves, driven by warming surface temperatures (Randall et al., 2020). Indeed, the sea surface temperatures in the Coral Sea in early 2024 were the warmest on reconstructed records spanning over 400 years (Henley et al., 2024). Furthermore, ENSO-related temperature and transport variability also impact the health of the GBR. The reduction in wind strength and storm activity during El Niño events can lead to increased short-wave radiation input and less mixing of the water column (Benthuysen et al., 2018; Berkelmans et al.,

2010). These conditions are conducive to the bleaching of corals, although an El Niño is not always a necessary prerequisite for mass coral bleaching (Hughes et al., 2018; McGowan and Theobald, 2023). In another example, interannual variability of the circulation, due to ENSO, has been linked to changes in GBR larval connectivity, important for coral recruitment (Gurdek-Bas et al., 2022). Understanding the relative strength and interplay between climate change and ENSO variability in the Coral Sea is key for the effective management of their ecosystems.

Most future climate projections are made with General Circulation Models (GCMs), with ocean components with resolutions of 1° latitude by 1° longitude (~111 km at the equator). The mesoscale processes important in the formation, transport and bifurcation of the jets moving in and around the Coral Sea cannot be captured at that resolution (Hewitt et al., 2020). In addition to the mesoscale processes, the blocking effects of the large islands in the Coral Sea are a first-order feature of the southwest Pacific that control the separation of the SEC jets, and the bathymetry and passages control the further separation of the jets

and, ultimately, their latitudinal position (see uncertainties in the island rule calculation in Kessler and Cravatte [2013b]). The capability to define coastlines and bathymetric obstacles in the complex Coral Sea is clearly limited in standard resolution GCMs (e.g., Figure S10, Sen Gupta et al., 2021).

The high-resolution coupled modelling experiment (HighResMIP) conducted within the latest iteration of the Coupled Model Intercomparison Project (Phase 6 [CMIP6, 2019] under the World Climate Research Programme [WCRP, 2024]) included

GCMs with ocean components of 1/4° resolution (nominal resolution 25 km) that could resolve mesoscale processes (Haarsma et al., 2016). The highest resolution HighResMIP models even had eddy permitting 1/10° (nominal resolution 10 km) ocean components (Bao and He, 2019; Hurrel et al., 2020). With simulations running from 1950 to 2050, the highest resolution models of HighResMIP provide a unique opportunity to investigate how the SEC jets in the Coral Sea may respond to climate change over the 100-year period, and to examine which climate signal may propagate towards the GBR.

The broad aim of this study was to investigate the historical representation and near future (2050) projections of the SEC jets in the Coral Sea, using the highest resolution models of the CMIP6 HighResMIP experiment. In section 2 we compared the global atmospheric warming from the selected CMIP6 HighResMIP models to the broader CMIP6 ensemble and to observations to provide context for the HighResMIP projections. The methods of the targeted Coral Sea and SEC analysis are outlined in section 3. In section 4, the representation of the modelled Coral Sea heat storage, SEC jet mean states and ENSO-
related interannual variability are assessed against the BRAN2020 ocean reanalysis (Chamberlain et al., 2021). In section 5, the projected near future trends in the SEC jets are analysed within the context of the magnitude of the historical ENSO-related variability. This investigation deepened our understanding of climate change impacts on the SEC and the wider Coral Sea, as discussed in section 6.

**2 HighResMIP models and global warming projections**

**2.1 HighResMIP model selection**

The spatial scale of the ocean dynamic processes in the Coral Sea region required that we restricted our analysis to HighResMIP models with resolutions of at least 1/4° (25 km nominal resolution) in the ocean and atmosphere. Four HighResMIP models met our resolution criteria, and three were retained for analysis based on data availability (Table 1). The HighResMIP model
from the Research Centre for Environmental Changes: Academia Sinica (AS-RCEC) Hiram-SIT-HR (Tu, 2020) was excluded as no ocean component was available for download at the time of writing.

Table 1. Details of the CMIP6 HighResMIP models used in this study. Hereafter, CESM1-CAM5-SE-HR, CMCC-CM2-VHR4, FGOALS-f3-H are abbreviated to CESM, CMCC and FGOALS, respectively.

| Model abbreviation | Full model name (modelling group) | Atmospheric model (resolution) | Oceanographic model (resolution) | Ocean vertical levels | Top ocean grid cell depth bounds | Reference |
|---|---|---|---|---|---|---|
| CESM | CESM1-CAM5-SE-HR (NCAR) | CAM5.2 (25 km) | POP2 (10 km) | 62 | 0 – 10 m | (Hurrel et al., 2020) |
| CMCC | CMCC-CM2-VHR4 (CMCC) | CAM4 (25 km) | NEMO3.6 (25 km) | 50 | 0 – 1 m | (Scoccimarro et al., 2017) |

| FGOALS | FGOALS-f3-H (CAS) | FAMIL2.2 (25 km) | LICOM3.0 (10 km) | 55 | 0 – 5 m | (Bao and He, 2019) |

## 2.2 Global warming projections

Comparing the HighResMIP model projections with the coarser CMIP6 projections was important for two reasons. Firstly, the HighResMIP experiment only included a high-forcing climate scenario, the Shared Socio-economic Pathway 5-8.5 (SSP5-8.5). Secondly, the HighResMIP was run from 1950 to 2050, as opposed to the standard 1850 to 2100 period.

The warming in the global averaged near surface air temperature from the 1850-1879 climatology is widely used to quantify global warming. In Figure 2, we compare this metric from the three HighResMIP models, the CMIP6 Scenario Model Intercomparison Project ensemble (ScenarioMIP; O'Neill et al., 2016) and observations. For each model, the warming trend was calculated from the area weighted global average near-surface air temperature. For the HighResMIP models, we could only track the warming trend from 1950 (the start of the HighResMIP experiment) and we set the starting point as the 1950

ScenarioMIP SSP5-8.5 ensemble mean. For the observations, three land ocean global surface temperature datasets were included for the historical period January 1850 to May 2024: the NOAA Global Surface Temperature Dataset (NOAAGlobalTemp; Huang et al., 2024), the Met Office Hadley Centre Observations Dataset (HadCUT5; Morice et al., 2021) and the Berkeley Earth Land/Ocean Temperature Record (Rohde and Hausfather, 2020). The observation time-series were averaged together (Figure 2).

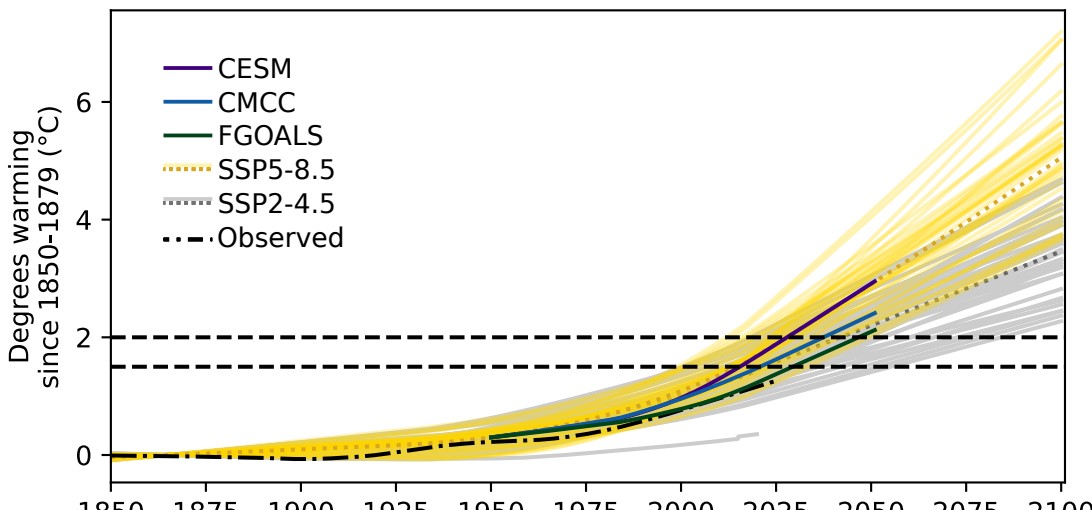

**Figure 2. Trend in degrees warming since the pre-industrial 1850 – 1879 historical period, calculated from the global average near-surface air temperature. Bold unbroken lines show the HighResMIP models, which were run for SSP5-8.5. For both the SSP5-8.5 (38 models) and SSP2-4.5 (37 models) ScenarioMIP ensembles, unbroken light lines are ensemble members and the dotted dark line**
**is the ensemble mean. The observed (dashed dotted line) is the average of the NOAA Global Surface Temperature Dataset, the Met**

Office Hadley Centre observations dataset and the Berkeley Earth Land/Ocean Temperature Record. Dashed lines indicate 1.5°C and 2°C warming.

The warming trajectories of the HighResMIP models were located in the middle of the range of the broader CMIP6 ScenarioMIP projections (Figure 2). Compared to the SSP2-4.5 ensemble, the HighResMIP models simulated moderate (FGOALS, CMCC) to high (CESM) warming of the atmosphere. Furthermore, the warming trajectory in FGOALS in the historical period tracked with the observed rate of warming, while the rates of warming in CESM and CMCC were faster than observed. For example, CESM and CMCC both prematurely forecast that 1.5°C warming would be exceeded before 2020, while it was not observed as of December 2023 (observed: 1.24°C; Table S1). While our analysis is limited to three models, the warming trajectories span almost half of the projected CMIP6 ensemble and provide an opportunity to explore how the global climate signal may propagate into the Coral Sea region in the near future.

## 3 Coral Sea data treatment

Herein we detail the methodology applied to the HighResMIP model data for the assessment of the Coral Sea and the SEC system. We first describe the removal of internal model simulation drift in ocean temperature and salinity. We then determined the spatial extent of the SEC jets, extracted them, and applied a temporal decomposition to evaluate the trends and interannual variability associated with El Niño and La Niña conditions.

### 3.1 Drift analysis and removal

It is common for the heat and salt budgets of oceans to drift internally when GCMs are initialised after standard 100 year spin-up periods (Irving et al., 2021). Internal drift was even more likely to occur in the HighResMIP models given their short spin-up times of 50 years. Control runs with the same spin-ups as scenario runs but with fixed climatological greenhouse gas levels post-spin-up can be used to de-drift scenario runs. Given the fixed atmosphere, any trends in ocean properties in the control runs can be attributed to internal model drift. Therefore, removing the control drift from scenario runs effectively balances the heat and salt budgets (e.g., Irving et al., 2021).

Here, we assessed and corrected the drift in heat and salt budgets in the HighResMIP models. We utilized the HighResMIP control runs and analysed the top 1000 m of the water column in the Coral Sea. The sea boundaries used for all Coral Sea-scale analyses in this paper were sourced from the International Hydrographic Organization (IHO) sea areas (Flanders Marine Institute, 2018; Figure 1a). The heat budget was assessed by calculating the Ocean Heat Content (OHC; J m$^{-3}$) of the Coral Sea as follows:

$$OHC = \frac{\rho C \iiint T V_C \, dx \, dy \, dz}{V_T} \tag{1}$$

With the variables: potential temperature ($T$; K), grid cell volume ($V_c$; m³), density ($\rho$; assumed constant; 1026 kg m⁻³), specific heat capacity ($C$; assumed constant = 4000 J kg⁻¹, K⁻¹) of sea water, and the total volume of the top 1000 m of the Coral Sea ($V_T$; m³). There was no long-term trend in the OHC in the CESM and FGOALS control runs. However, it decreased through time in the CMCC control run (Figure S1a, c, e). Following the de-drifting procedure that Irving et al. (2021) used to balance the heat and salt budgets of CMIP5 and CMIP6 models, we used least squares regression to identify the grid cell-wise linear decrease in temperature through time that was present in the CMCC control run, and removed it from the SSP5-8.5 run. This effectively removed the change in temperature associated with internal drift.

Drift in the salt budget was assessed through analysing the change in Coral Sea salt content. The average salinity of the Coral Sea decreased through time in each of the HighResMIP model control runs (Figure S1b, d, f). The change in salinity associated with internal drift in the control runs was removed from the SSP5-8.5 runs following Irving et al. (2021)'s method.

## 3.2 Spatial extraction of jet data

We focused our analysis on the jets of the South Equatorial Current that enter the Coral Sea (Figure 1a). As the HighResMIP model data are available as monthly means, eddy activity is smoothed from the datasets. Model-specific meridional slices of the North Vanuatu Jet (NVJ), the North Caledonian Jet (NCJ) and the South Caledonian Jet (SCJ) were extracted at a longitude of 162°E for each of the HighResMIP models and the BRAN2020 ocean reanalysis (Figure 1a, Figure S2a). The slices were restricted to the top 1000 m of the water column, to capture the majority of the transport of the surface-intensified jets. The minimum and maximum latitudes of the jets were selected from Hovmöller diagrams of the zonal current speeds on the 162°E meridian, from which the jets were clearly visible. Slices of the monthly average zonal current speed, temperature and salinity were extracted.

As the jet positions vary in time, the jet areas were defined as the area within each zonal current speed slice where the current flowed in the prevailing westward direction. Thereby, time-variant masks were applied to extract the jets characteristics from the data. We focused on the predominant westward portion of the flow as it transports the climate signal through the Coral Sea to the Australian continental shelf break and into western boundary currents. The masked data were finally meridionally averaged to get jet-specific depth profiles through time. The time-variant masks were also used to generate zonal volume transport profiles ($U$; m³ s⁻¹) as follows:

$$U = \int u\,a\,dy \tag{2}$$

With the westward current speed ($u$; m s⁻¹) and the grid cell area ($a$; m²) over the jet-specific masked area.

## 3.3 Temporal decomposition of jet data

The data were decomposed into seasonal and interannual components and a long term trend such that:

$$u = u_{season}' + u_{ia}' + u_{trend} + \varepsilon \tag{3}$$

Where a seasonal decomposition was performed (python 3.10.13, statsmodels version 0.14.1, STL function with a set period of 12 months; Cleveland et al., 1990) to separate the data into seasonal anomalies (e.g., $u_{season}'$), the interannual variability

with the trend, and a residual $\varepsilon$. We then performed Locally Weighted Scatterplot Smoothing (LOWESS) on the combined interannual variability and trend to extract the trend component (e.g., $u_{trend}$; python 3.10.13, statsmodels version 0.14.1, lowess function; Cleveland, 1979). We calculated interannual anomalies (e.g., $u_{ia}'$) by subtracting the trend from the interannual variability.

An additional analysis was conducted on the interannual components to identify variability associated with the El Niño

Southern Oscillation (ENSO). Firstly, the Niño 3.4 index was calculated for each of the HighResMIP models. The warming trend was removed from the temperature data before calculating the index. The observed Niño 3.4 index calculated from the Extended Reconstructed Sea Surface Temperature (ERSST) dataset from the National Oceanographic and Atmospheric Administration (NOAA, Monthly Atmospheric and SST Indices, 2024) was used to determine the timing of El Niño and La Niña conditions in BRAN2020. El Niño and La Niña conditions were identified as when the Niño 3.4 index exceeded 0.4, or

was below -0.4, respectively (Trenberth, 1997).

Secondly, the normalised interannual components were cross-correlated with the normalised Niño 3.4 indices at each depth to determine model/current specific lagged effects of the ENSO forcing on the variables (Figures S3 to S5; python 3.10.13, numpy version 1.26.4, correlate function). The cross-correlation was done using all available model data, from 1950 to 2050 for the HighResMIP models and from 1993 to 2023 for BRAN2020, to capture as many El Niño and La Niña events as possible.

The lags corresponding to the highest correlations were identified for each variable/model/current combination. Finally, the interannual components were extracted during the identified El Niño and La Niña conditions plus the relevant lags to generate El Niño and La Niña composite anomalies.

**3.4 Heat transport, depth, and stratification**

Additional calculations were carried out on the trend components. The heat transported ($Q_{trend}$; J s$^{-1}$) by the SEC jets within the top 1000 m of the water column was calculated as follows:

$$Q_{trend} = \rho C \int_0^{1000} T_{trend} \, U_{trend} \, dz \qquad (4)$$

Using the temperature ($T_{trend}$; K) and zonal volume transport ($U_{trend}$) trend components. We performed additional calculations with the NVJ data to elucidate which quantity (temperature, transport or both) was driving the heat transport trend.

Specifically, we calculated the heat transport using 1) the zonal volume transport trend and climatological temperature ($\overline{T}_{trend}^{1993-2023}$) averaged over the 1993 – 2023 period and 2) the climatological zonal volume transport ($\overline{U}_{trend}^{1993-2023}$) and temperature trend.

As the NCJ has a strong sub-surface transport maximum, we assessed if the depth of the maximum was stable through time in the HighResMIP models. For each time-step, we identified local maxima and minima in the zonal volume transport trend

vertical profiles. We recorded the depth of the identified dominant maximum.

Lastly, the stratification of the current jets through time was investigated. Brunt-Vaisala frequencies ($N^2$) were calculated from the temperature and salinity trend profiles using the Gibbs SeaWater Oceanographic Toolbox (gsw package, version 3.6.17). The maximum Brunt-Vaisala frequency was calculated for each time step and the depth of the maximum at each time step was taken as the mixed layer depth. These calculations, in conjunction with the temporally decomposed data, were used to

investigate the historical representation and near future projections of the Coral Sea and SEC current jets in the HighResMIP models.

## 4 Historical assessment

Before looking at what the HighResMIP models project for the future, we assess how well the models represented the complex hydrodynamics of the Coral Sea during the historical period. The HighResMIP models were benchmarked against the

BRAN2020 global ocean model reanalysis (Chamberlain et al., 2021b). BRAN2020 assimilates a large number of observational datasets, including satellite sea surface temperatures, satellite sea level anomalies and in-situ temperature and salinity (Chamberlain et al., 2021b). The broad-scale ocean features and the mesoscale features are constrained separately in a two-step data assimilation that delivers improved accuracy (Chamberlain et al., 2021a). BRAN2020 was chosen for use in this study because 1) it was developed with a focus on Australia and the surrounding region and (Chamberlain et al., 2021b)

2) it has been used to provide ocean boundary conditions for a regional model of the Great Barrier Reef, which requires skilful representation of the South Equatorial Current jets (Maggiorano et al., 2025). In the historical assessment, we specifically focused on the 31-year time period from 1993 to 2023 for which BRAN2020 data were available.

### 4.1 Temperature structure and OHC

Within the Coral Sea, the temperature structures of the SEC jets were well represented in the HighResMIP models. In BRAN2020, the NVJ, NCJ and SCJ had sea surface temperatures (SSTs) of 28.5°C, 27°C and 24°C, respectively, and the HighResMIP models captured this latitudinal gradient (Figure 3; Table 2). Furthermore, the HighResMIP models captured the vertical structure of the temperature profiles as in BRAN2020, where the mixed layers within all the current jets were relatively shallow (< 100 m; Figure S6) and the bottom depth of the thermocline increased from approximately 400 m to 600 m with

increasing latitude. While the HighResMIP model temperature profiles broadly aligned with BRAN2020, CESM and CMCC consistently decreased in temperature slightly too quickly with depth, and FGOALS decreased in temperature slightly too slowly. Notably, the mixed layers and thermocline areas of the BRAN2020 and HighResMIP profiles were similar to the

observed climatological Southern Hemisphere subtropical (15°S – 25°S) temperature profile (Figure 3.5 in Ganachaud et al. 2011).

The differences in vertical temperature distribution influenced the OHC, where higher temperatures in the top 200 m of the Coral Sea resulted in higher OHCs (Figure 1b). However, the magnitudes of the differences were relatively small; the mean OHCs in the HighResMIP models were within ±0.2% of the mean OHC in BRAN2020, indicating the HighResMIP models reasonably represented the best estimate of the Coral Sea OHC state. In summary, the HighResMIP models captured both the SEC jet temperature structures and the OHC of the Coral Sea.

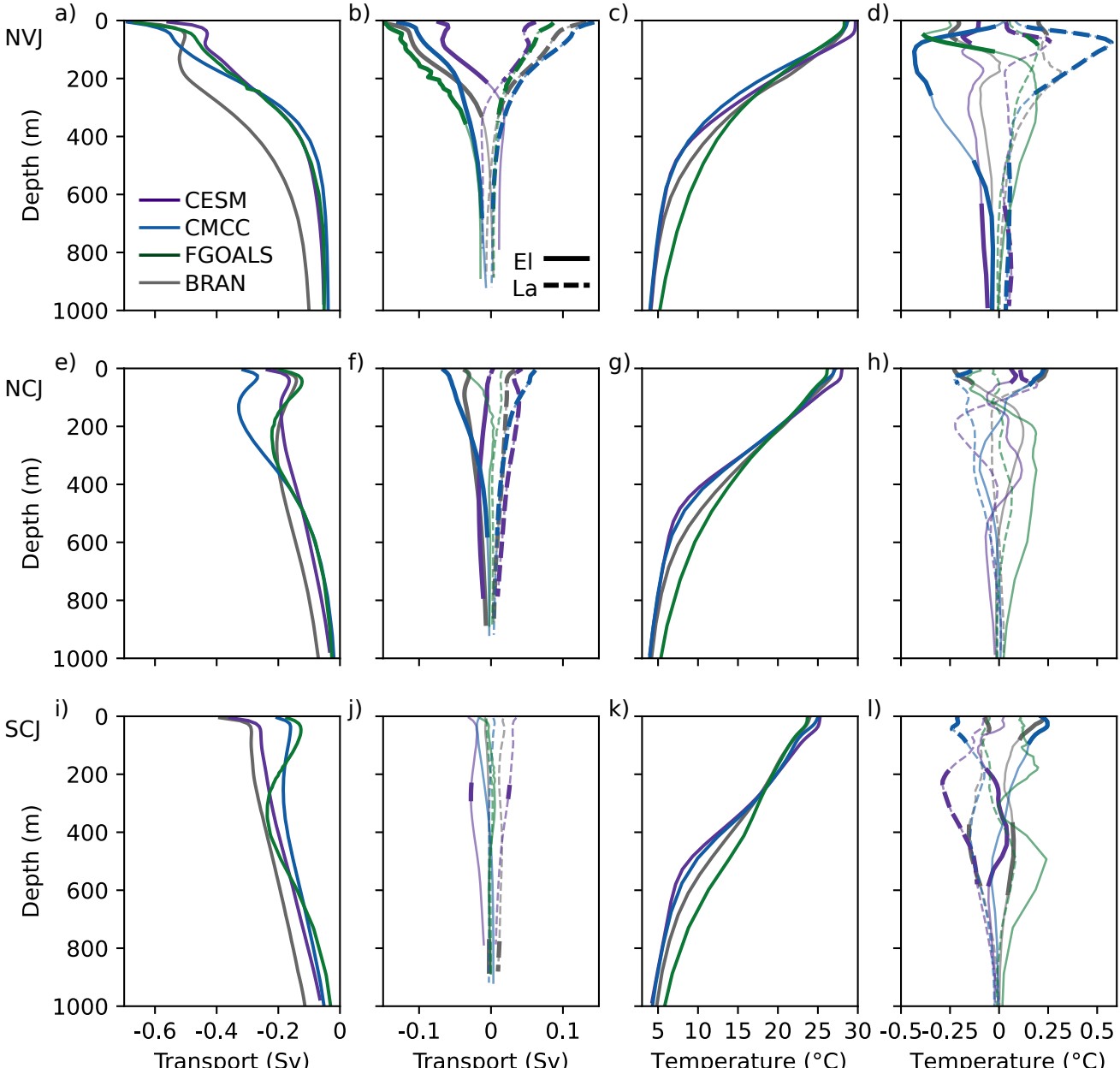

**Figure 3. South Equatorial Current jet historical assessment (1993 – 2023). Row 1: NVJ. Row 2: NCJ. Row 3: SCJ. Column 1: Climatological volume transport. Column 2: Volume transport anomalies associated with El Niño (El; solid lines) and La Niña (La; dashed lines) conditions. For transport, positive is eastward. The lines are bold at depths where the correlation ≥ 0.4, calculated from the cross-correlation profiles cut at the maximum correlation. Column 3: Climatological temperature. Column 4: El Niño and La Niña temperature anomalies. For visualization, the volume transport profiles were calculated over standardized 10 m depth bins.**

Table 2. Change in the 0 to 1000 m transport and SST of the NVJ, the NCJ and the SCJ with the ENSO. The climatological

means ± standard deviations (SD) from 1993 to 2023 when BRAN2020 data were available are shown. The changes were

calculated from the El Niño and La Niña composite anomalies and are expressed as absolute values and percentages relative

to the climatological means. The sign of change with El Niño is shown in the table, and La Niña is the opposite sign.

| | | Mean 1000 m transport. 1993 to 2023. Sv ± SD | Change with El Niño. ± Sv (%) | Mean SST. 1993 to 2023. °C ± SD | Change with El Niño. ± °C (%) |
|---|---|---|---|---|---|
| NVJ | BRAN2020 | 27.1 ± 1.0 | + 2.4 (9%) | 28.5 ± 0.2 | - 0.20 (0.7%) |
| | CESM | 17.0 ± 0.2 | + 1.9 (11%) | 29.7 ± 0.2 | - 0.07 (0.2%) |
| | CMCC | 17.2 ± 0.1 | + 2.8 (16%) | 28.8 ± 0.1 | - 0.06 (0.2%) |
| | FGOALS | 17.4 ± 0.2 | + 2.9 (17%) | 28.3 ± 0.1 | - 0.15 (0.5%) |
| | | | | | |
| NCJ | BRAN2020 | 14.9 ± 0.6 | + 1.5 (10%) | 27.0 ± 0.2 | - 0.24 (0.9%) |
| | CESM | 12.0 ± 0.2 | + 1.6 (13%) | 28.0 ± 0.2 | - 0.09 (0.3%) |
| | CMCC | 15.1 ± 0.1 | + 1.6 (11%) | 27.2 ± 0.2 | - 0.18 (0.7%) |
| | FGOALS | 11.6 ± 0.1 | NA | 26.2 ± 0.2 | NA |
| | | | | | |
| SCJ | BRAN2020 | 21.5 ± 0.8 | NA | 24.0 ± 0.2 | + 0.15 (0.6%) |
| | CESM | 17.3 ± 0.1 | NA | 25.3 ± 0.2 | + 0.05 (0.2%) |
| | CMCC | 13.6 ± 0.3 | NA | 25.1 ± 0.2 | + 0.21 (0.9%) |
| | FGOALS | 14.2 ± 0.05 | NA | 23.7 ± 0.2 | NA |

## 4.2 SEC jets transport and heat transport

As in BRAN2020, all HighResMIP models captured the topographic steering of the SEC into the NVJ, NCJ and SCJ as it

entered the Coral Sea (Figure 4, Figure S2). However, there were model-specific differences in the volume transports of the

jets (Table 2). This section compares the structure of the modelled jet transports with literature estimates.

### 4.2.1 NVJ

In BRAN2020, the NVJ was represented as a wide current that transported water westward between 11°S and 15.5°S south of the Solomon Islands. There was a surface transport maximum with a velocity of 0.16 m s$^{-1}$, and a sub-surface maximum of 0.12 m s$^{-1}$, below which the velocity decreased to 0.04 m s$^{-1}$ (Figure 3, Figure S7). Broadly, the geographical location, jet width in the first 200m, surface speed and transport of the NVJ in all three HighResMIP models were similar to BRAN2020. However, the speed and transport reduction with depth diverged from BRAN2020, with a faster decrease with depth in the HighResMIP models, resulting in large errors below 200 m (Figure 3, Figure 4). In particular, the width of the NVJ in CESM and CMCC narrowed significantly below 400 m relative to BRAN2020. The NVJ in FGOALS also had an atypical, pronounced transport maximum within the jet at approximately 12°S. These disparities likely contributed to differences in depth-integrated transport (Table 2), with the NVJ in BRAN2020 transporting 27 Sv above 1000 m and the HighResMIP models transporting 17 Sv. Notably, Ganachaud et al. (2014) reported a modelled and observed NVJ transport of 20 Sv which more closely aligned with estimates from the HighResMIP models. However, the structure of the NVJ in BRAN2020 (Figure 4a) better matched the structure reported from geostrophic velocities in Figure 8a from Ganachaud et al. (2014) and Figure 13g from Kessler and Cravatte (2013b), where the 0.1 m s$^{-1}$ isotach extended down to approximately 250 m. Heat transport differences were essentially dominated by transport differences, where variance in temperature contributed little to the overall variance in heat transport (Figure S8) and, consequently, the magnitudes of heat transported by the NVJ over 1000 m in the HighResMIP models were also two-thirds the magnitude calculated for BRAN2020 (Figure S9).

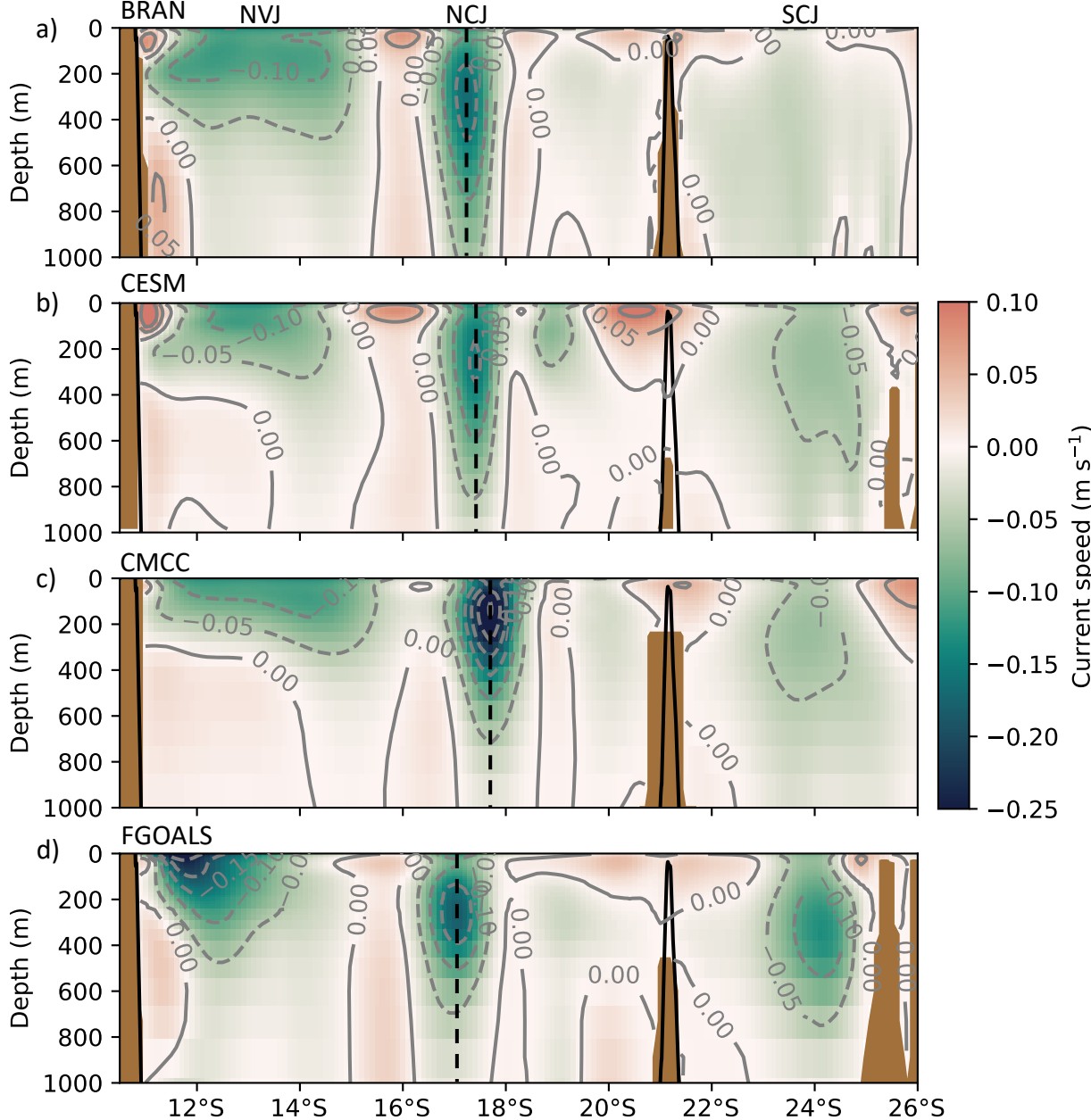

**Figure 4. South Equatorial Current jet historical assessment (1993 to 2023). Mean current speed meridional 162°E sections for a) BRAN2020 and b) CESM, c) CMCC and d) FGOALS. Positive is eastward. The positions of the NVJ, the NCJ and the SCJ are indicated. The black dashed lines show the latitude of maximum NCJ current speed. The model bathymetry is shown in brown, and the GEBCO 2023 bathymetry is overlayed as solid black lines (GEBCO, 2024). Contours are in grey; positive contours and solid and**
320 **negative contours are dashed.**

### 4.2.2 NCJ

There was generally closer alignment of the HighResMIP models with BRAN2020 in their representation of the NCJ (Figure 3, Figure 4), in comparison with that of the NVJ. The NCJ in BRAN2020 was narrower than the NVJ, transporting water westward between 16°S and 18°S at 162°E. In BRAN2020, the NCJ had two local westward transport maxima: one at the surface and one at 285 m, with respective velocities of 0.12 m s$^{-1}$ and 0.11 m s$^{-1}$ (Figure S7). The geostrophic component of the NVJ in Ganachaud et al. (2014) also had surface and subsurface transport maxima (their Figure 8a), while only the subsurface maximum was present in the geostrophic velocities in Kessler and Cravatte (2013b, their Figure 13h). The two maxima were captured in CESM and FGOALS; however, CMCC displayed an overly strong velocity maximum at 180 m depth and was displaced to the south by 0.24° compared with the NCJ centre in BRAN2020 (Figure 4c). Moreover, the NCJ in FGOALS was shifted 0.18° north in comparison to BRAN2020. The NCJ transport of 12 Sv in CESM and FGOALS underestimated by 25% the BRAN2020 transport of 15 Sv (Table 2). In CMCC, the overestimated subsurface transport balanced the underestimated deep transport so the transport over 1000 m approximately matched BRAN2020. The BRAN2020 and HighResMIP model mean NCJ transports were approximately aligned with the calculated geostrophic and modelled transport of 12 – 13 Sv reported in the literature (Ganachaud et al., 2014; Kessler & Cravatte, 2013b). The comparison of heat transport in the HighResMIP models versus BRAN2020 mirrored the transport comparison, where the NCJ in CESM and FGOALS transported slightly less heat, and the heat transported by the NCJ in CMCC fell within the range of BRAN2020 values (Figure S9).

### 4.2.3 SCJ

Broadly, the SCJ in the HighResMIP models transported less volume compared to the BRAN2020 representation (Figure 3, Figure 4). Geostrophic calculations of the SCJ suggest that it is a subsurface current that travels south of New Caledonia (Ganachaud et al., 2008; Ganachaud et al., 2014; Kessler & Cravatte, 2013b). The SCJ in BRAN2020 was a wide current that transported water westward between 21°S and 26°S and was surface intensified, in contrast to the geostrophic calculations from the literature, with a surface velocity of 0.12 m s$^{-1}$ (Figure S7). FGOALS was the only model to have a subsurface SCJ in line with literature reports. Notably, BRAN2020 and the HighResMIP models would carry Ekman components that are not included in the geostrophic component of the SCJ reported in the literature, which could contribute to the difference in vertical structure. The SCJ transported 22 Sv over 1000 m in BRAN2020, and 14 to 17 Sv in the HighResMIP models, representing 60 to 80% of the BRAN2020 transport. Notably the HighResMIP model transports fell within the upper range of the 0 to 1000 m SCJ transports calculated across a general ocean circulation model ensemble in Cravatte et al. (2021), and the BRAN2020 SCJ transport exceeded the reported range. However, Cravatte et al. (2021) found their ensemble under-estimated the SCJ transport relative to Argo data. The representation of the Capel-Faust Seamounts on the Lord Howe Rise (see Figure 1 in Gilmore et al., 2020) in the model bathymetries appears to greatly impact their representation of the transport and shape

of the SCJ. For example, the seamounts were represented as prominent features north of 26°S on the 162°E meridian in CESM and FGOALS and were much less prominent in the GEBCO 2023 bathymetry, BRAN2020 or CMCC (Figure 4, Figure S10). In CESM and FGOALS, the seamounts obstructed the SCJ resulting in a narrower jet compared to BRAN2020 and CMCC. Finally, as for the NVJ and the NCJ, disparities in the heat transport reflected the transport differences where the SCJs in the HighResMIP models transported less heat compared to BRAN2020 (Figure S9).

## 4.3 ENSO variability

The responses of the SEC jet transport to ENSO in BRAN2020 differed to the North and South of New Caledonia (Figure 3, Figure S7). North of New Caledonia, the NVJ and NCJ showed lagged 9% and 10% increases in strength in response to El Niños, and decreases following La Niñas (Table 2). In contrast, interannual variability in the SCJ transport was not correlated with ENSO. These ENSO relationships concur with the observed increases/decreases in transport into the Coral Sea by the NVJ and NCJ in the months following El Niño/La Niña events (Kessler and Cravatte, 2013a). The temperature of the SEC jets varied with ENSO, where the depth and direction of the response was dependent on the jet (statistically significant correlation ≥ 0.4; Figure 3, column 4). However, the magnitude of the ENSO-related temperature variability was small (< 1%) relative to the mean temperatures.

The HighResMIP models broadly captured the ENSO related variability of the NVJ and NCJ. The transport responses were similar to BRAN2020 in CESM and CMCC but not in FGOALS, which only captured the correlations and monthly lags of the NVJ response (Figure 3, Table 2, Figure S11). Furthermore, the monthly lags of the correlations between temperature and Niño 3.4, in CESM and CMCC were similar to the correlation lags in BRAN2020, although there was some deviation in the lag times corresponding to the maximum correlations.

Disparities between the models might come from a different representation of the ENSO oscillation. There was a greater range in peaks of the Niño 3.4 index associated with El Niño and La Niña events in the observations (1950 to 2023) compared to the HighResMIP models (1950 – 2050; Figure S12). However, this diversity did not translate to differences in the magnitude or direction of the responses of SEC jet temperature and transport to ENSO.

## 5 Projected 2050 changes

The near future changes the HighResMIP models projected for the Coral Sea and SEC jets, relative to the 30-year 1950-1979 historical reference period, are described in this section. To account for the uncertainties inherent in climate projections, we averaged anomalies across the three HighResMIP models. Model-specific anomalies are available in the supporting information.

## 5.1 Temperature and OHC trend

The HighResMIPs models projected an ensemble mean maximum warming of 1.4°C ± 0.2°C standard deviation (SD) of the SST in the Coral Sea (Figure 5) between 1950 and 2050. The SST trend was not constant through time, where the rate of warming doubled from 0.09°C per decade ± 0.02°C SD prior to 1985 to 0.21°C per decade ± 0.02°C afterward. Furthermore, the warming signal deepened by approximately 30 m per decade, extending to 400 m below the surface by 2050. The depth structure of the temperature changes were relatively consistent across the HighResMIP models (Figure 5 and Figure S13) and were associated with an increase in Coral Sea OHC in the first 200 m (Figure 1b). There was cooling in waters underneath the warming band. However, the cooling signal was much smaller than the projected warming, with a mean minimum cooling of -0.2°C ± 0.05°C SD. This structure of surface/subsurface warming with a band of cooling underneath has also been reported for the tropical Pacific region to the north of the Coral Sea (Jiang et al., 2024; Ju et al., 2022).

Table 3. Projected change in the 0 to 1000 m transport and SST of the NVJ, the NCJ and the SCJ in 2050, relative to the 1950 – 1979 climatologies. The climatological means ± standard deviations (SD) are shown. The changes are expressed as absolute values and percentages relative to the climatological means.

| | | Mean 1000 m transport. 1950 to 1979. Sv ± SD | Change in 2050. ± Sv (%) | Mean SST. 1950 to 1979. °C ± SD | Change in 2050. ± °C (%) |
|---|---|---|---|---|---|
| NVJ | CESM | 17.6 ± 0.1 | - 0.9 (5%) | 29.0 ± 0.1 | 1.9 + (6%) |
| | CMCC | 17.6 ± 0.05 | - 0.5 (3%) | 28.3 ± 0.1 | 1.3 + (5%) |
| | FGOALS | 17.6 ± 0.2 | + 0.4 (2%) | 27.8 ± 0.1 | 1.2 + (4%) |
| | | | | | |
| NCJ | CESM | 12.8 ± 0.1 | - 1.2 (10%) | 27.3 ± 0.1 | 1.8 + (7%) |
| | CMCC | 15.5 ± 0.2 | - 0.2 (1%) | 26.7 ± 0.1 | 1.4 + (5%) |
| | FGOALS | 11.9 ± 0.01 | - 0.9 (7%) | 25.7 ± 0.06 | 1.4 + (5%) |
| | | | | | |
| SCJ | CESM | 17.1 ± 0.05 | + 0.4 (2%) | 24.5 ± 0.1 | 1.8 + (7%) |
| | CMCC | 12.7 ± 0.1 | + 2.3 (18%) | 24.5 ± 0.1 | 1.5 + (6%) |
| | FGOALS | 14.3 ± 0.1 | + 0.01 (0.05%) | 23.2 ± 0.1 | 1.3 + (6%) |

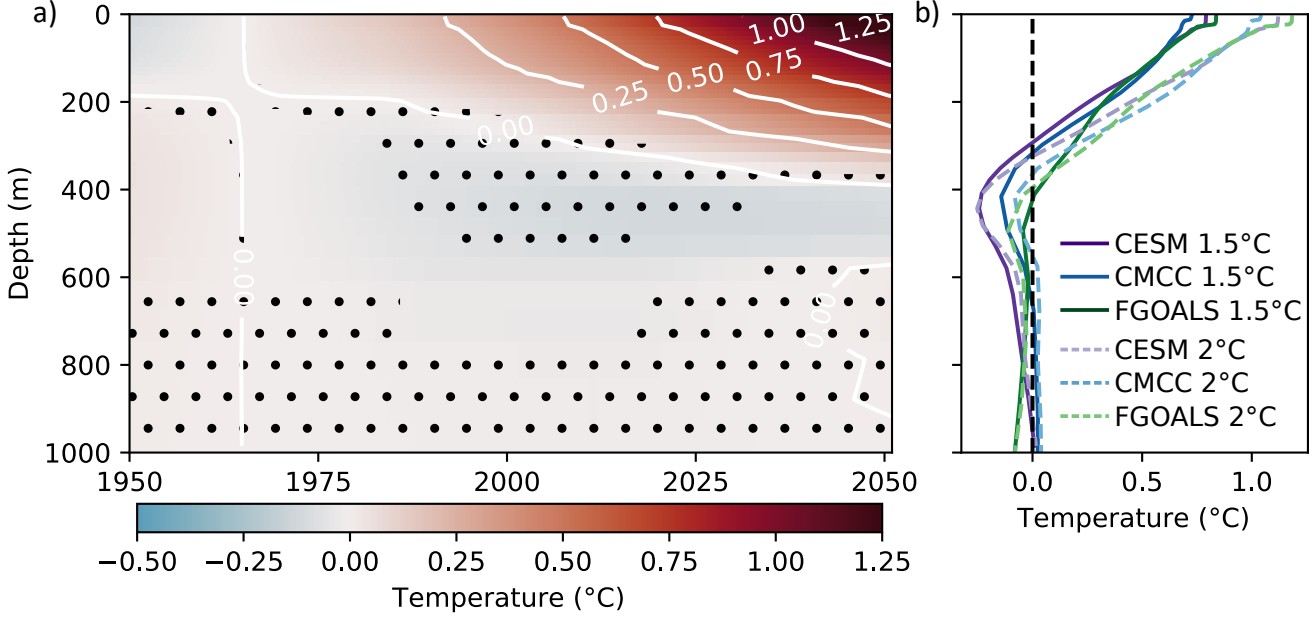

**Figure 5. Warming in the Coral Sea. a) HighResMIP model ensemble mean Coral Sea temperature anomaly (relative to 1950 – 1979). Contours are in white; positive contours and solid and negative contours are dashed. The stippling indicates depths/times when all three models did not agree on the sign of the anomaly. b) Temperature anomaly profiles under 1.5° C and 2° C global warming. A different time point was selected for each HighResMIP model, depending on when the model reached 1.5° C and 2° C**

**global warming (Table S1).**

Looking at projected changes under the 1.5°C and 2°C global warming thresholds, there were marked differences in the projected heating in the Coral Sea under 1.5°C versus 2°C global warming, with relatively consistent projections in the top 200 m of the water column across the three HighResMIP models (Figure 5b). The SST warmed from 0.78°C ± 0.04°C SD at

410 1.5°C global warming to 1.12°C ± 0.06°C SD at 2°C. The heating signal in the top 200 m under 1.5°C global warming was projected to deepen by approximately 80 m under 2°C of warming. There was more uncertainty in the trend direction below 200 m. Subsurface cooling was projected in all three HighResMIP models under 1.5°C of global warming; however, the magnitude varied from a maximum of -0.24°C at 400 m for CESM, to a maximum -0.10°C at 500 m for FGOALS. Under 2°C global warming, the magnitude of cooling was projected to remain the same for CESM, decrease by half for CMCC (relative

warming), and increase for FGOALS (relative cooling).

Looking at the temperature structure of the SEC jets, the pattern of temperature change simulated in the Coral Sea was replicated in the NVJ, NCJ and SCJ in all three HighResMIP models (Figure 6; Figures S14 to S16). The projected changes in temperature structure had a minimal effect on the mixed layer depths of the jets (Figure S6). However, there were consistent changes in the strength of stratification of the jets, with an average projected rate of increase of 1.0% decade$^{-1}$ ± 0.6% decade$^{-}$

$^1$ SD, relative to 1950 to 1979 climatologies. The magnitude of this change is similar to the rate of increase in stratification of 0.9% decade$^{-1}$ that has been observed throughout the world's oceans (Li et al., 2020).

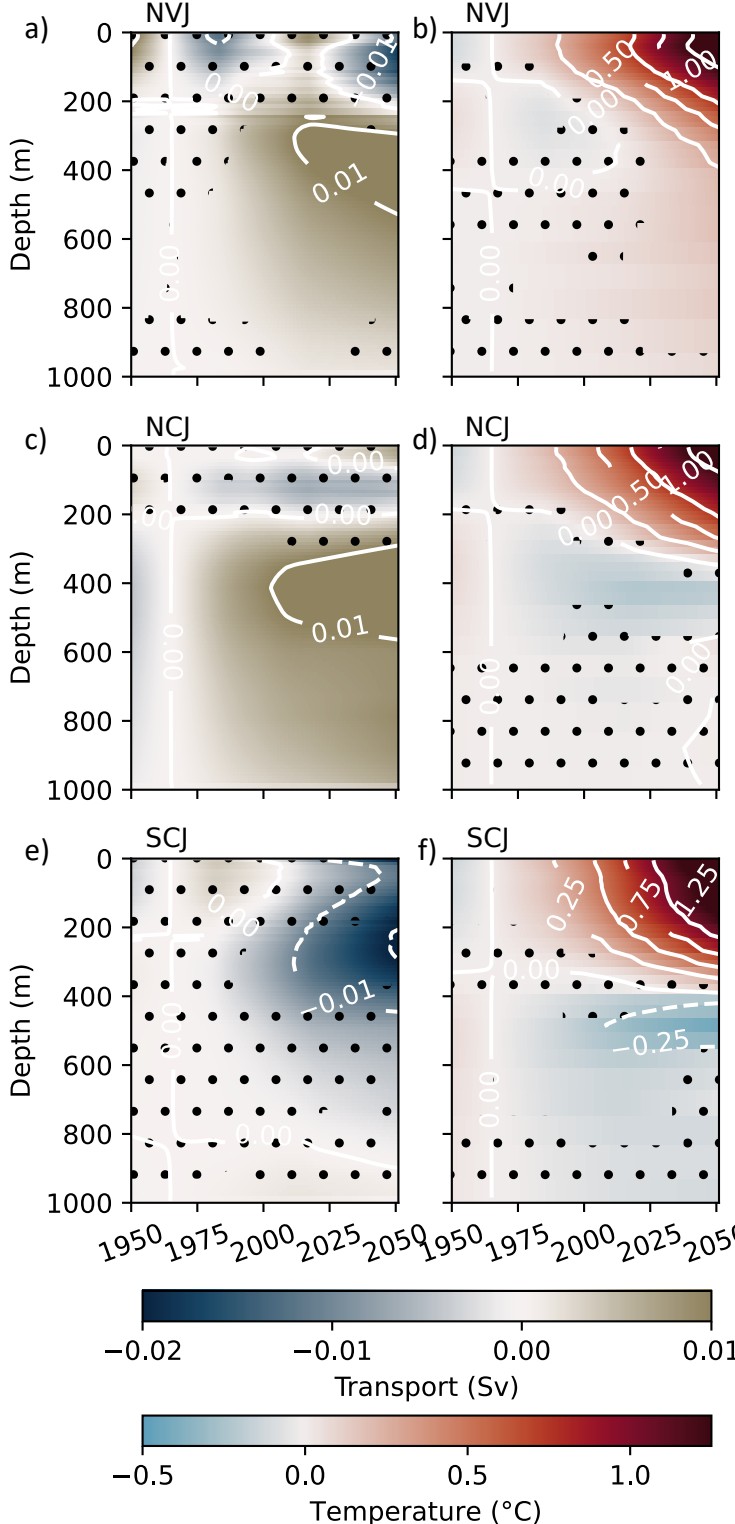

**Figure 6. Projected change in the South Equatorial Current jets (relative to 1950 – 1979). Row 1: NVJ. Row 2: NCJ. Row 3: SCJ. Column 1: HighResMIP model mean volume transport anomaly. For transport, positive (brown) is a decrease westward and negative (blue) is an increase westward. For visualization, the volume transport anomalies were calculated over standardized 10 m depth bins. Column 2: HighResMIP model mean temperature anomaly. Contours are in white; positive contours and solid and negative contours are dashed. The stippling on the mean anomalies indicates depths/times when all three models did not agree on the sign of the anomaly.**

## 5.2 Changes in transport and heat transport

Consistent changes in the transport of the SEC current jets were projected across the HighResMIP models. Overall, the 1000 m volume transports of the NVJ and NCJ were projected to weaken by 1.9% ± 3.1 SD and 6.1% ± 1.3 SD, respectively, and the transport of the SCJ was projected to strengthen by 6.8% ± 2.7 SD (Figure 6, Table 3). These changes were already established at 1.5°C global warming, and the magnitudes and signs of the changes remained similar at 2°C warming (Figure S17). The magnitudes of the projected decreases in transport in the NCJ and NVJ (1 to 10%, Table 3) were similar to the magnitudes of the increases/decreases associated with El Niño/La Niña events (9 to 17%, Table 2). Furthermore, the decreases in NVJ and NCJ transport align with the projected 18% decrease in SEC transport by 2100 under a high forcing scenario reported by Ganachaud et al. (2011). Notably, the SCJ strengthening was more prominent in CMCC, with only small intensifications in CESM and FGOALS (Figure 6e; Figure S16).

While the depth integrated transport weakened, near surface intensifications of NCJ and NVJ transport were simulated in two of the three HighResMIP models (Figure 7). These changes were also associated with a minor shallowing (20 – 50 m) of the subsurface maxima of the NCJ (Figure 7 e, g). Interestingly, the near-surface current intensifications overlapped with the depths that had the most pronounced warming. The corresponding strengthening and warming align with the expected intensification of surface currents as surface waters warm with climate change (Peng et al. 2022), and Ganachaud et al. (2011) also reported a surface strengthening overlying a weakening SEC. In Figure 7, we examined the changes in heat transport projections associated with the near-surface volume transport intensifications, by splitting the water column into two sections for the heat transport calculations, based on the depths of changes in the volume transports. The sections were: 1) a zone including the intensification, from the bottom of the intensification to the surface and 2) the water column below the intensification. The depth-varying volume transport and temperature changes were reflected in the heat transport projections, where the heat transport increased or was stable near the surface where the volume transport was projected to increase, and the heat transport decreased in the water column below, where the transport was projected to decrease (Figure 7 i, j).

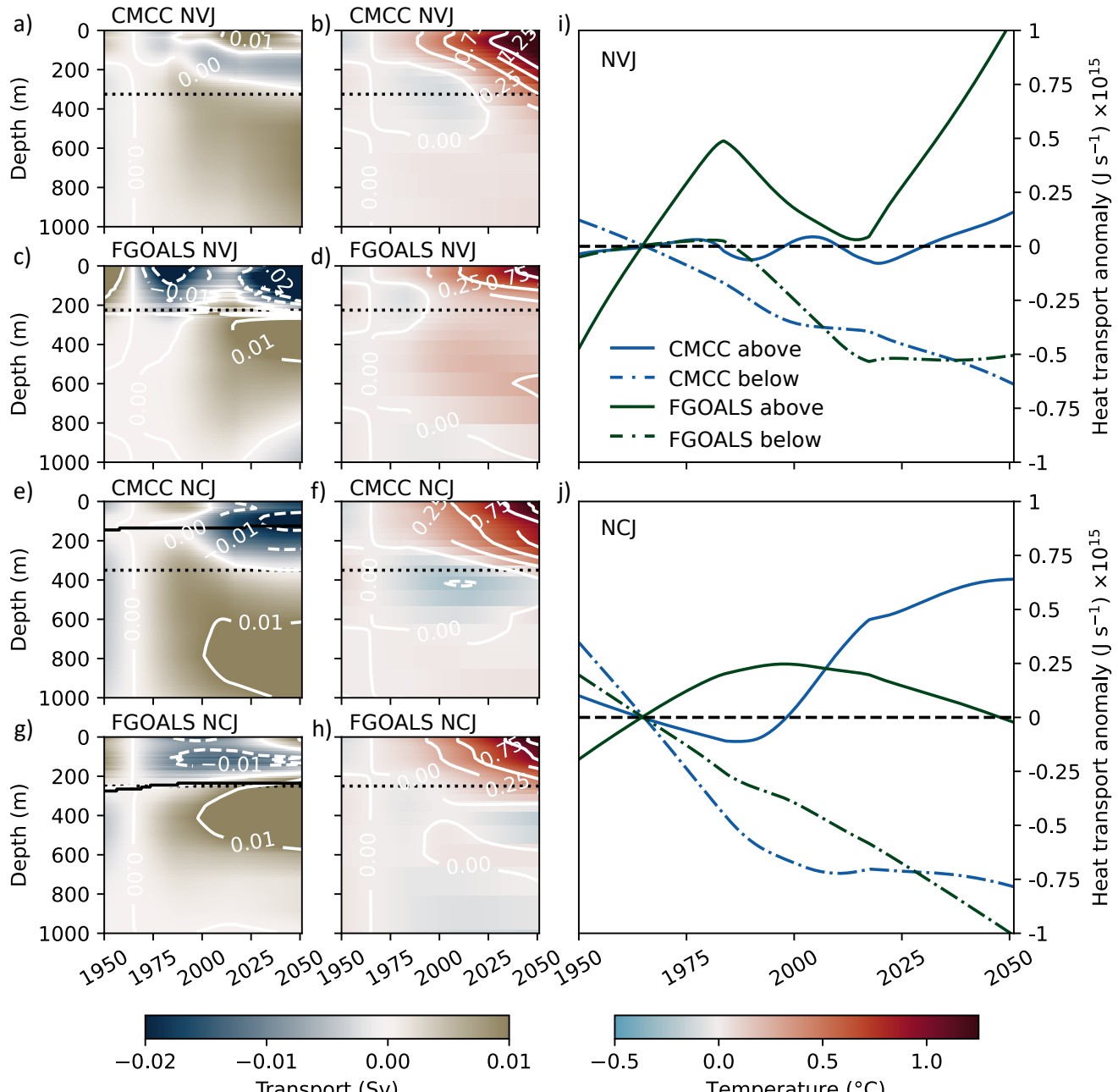

**Figure 7. Depth-resolved heat transport anomalies (relative to 1950 – 1979). a - d) NVJ volume transport and temperature anomalies. e - h) NCJ volume transport and temperature anomalies. i - j) NVJ and NCJ heat transport anomalies. For transport, positive (brown) is a decrease westward and negative (blue) is an increase westward. The volume transports are visualized over standardized 10 m depth bins. Contours are in white; positive contours and solid and negative contours are dashed. Dashed black lines on a-h indicate the depth where the water column was divided for the heat transport calculation. The solid lines on i-j show the anomalies above the division, and the dot dashed lines show the anomalies below. The solid black lines on e and g indicate the depths of the maximum NCJ subsurface transport through time.**

# 6 Discussion

## 6.1 The benefits of HighResMIP models

CMIP6 was developed to understand the long-time scale responses of the climate system to changes in radiative forcing (O'Neill et al., 2016). While this has facilitated an ever-increasing understanding of the impacts of climate change on the global ocean, CMIP6 experiments generally lacked the necessary resolution and process representation to provide reliable information at the continental shelf and coastal scale (Hewitt et al., 2020). However, the HighResMIP experiment included models with component resolutions up to 1/4° (mesoscale resolving) and 1/10° (eddy resolving) that could represent the complex SEC system. Thus, the HighResMIP provided the opportunity to understand regional changes in circulation in the Coral Sea.

We only utilized three HighResMIP models given the limited number of HighResMIP participants and our resolution-based selection criteria. However, the small ensemble size enabled us to comprehensively investigate each model. Two additional limitations of the HighResMIP are that the experiment only spanned the 1950-2050 time period and only included the high range emission shared socioeconomic pathway (SSP5-8.5), which has been deemed unlikely (Burgess et al., 2021; Hausfather and Peters, 2020). For the Coral Sea region, the 2050 timeline aligns well with local management initiatives (e.g., The Reef 2050 Long-term Sustainability Plan [Commonwealth of Australia, 2023]). Moreover, we demonstrated that the 2050 global warming projections of the three HighResMIP models used in our study aligned with the moderate to high range of the more likely "middle of the road" ScenarioMIP SSP2-4.5 ensemble. For a near-future outlook, the range of projections covered by the HighResMIP models may be particularly useful given the undershooting of low-range SSP2-4.5 projected warming after 2000 compared to observations (Figure 2). Overall, the HighResMIP models used in this study were of great utility given they were of sufficient resolution to simulate the circulation in the Coral Sea, provided a management-relevant near-future outlook and had global warming trajectories that conformed with the SSP2-4.5 ensemble.

## 6.2 HighResMIP model performance

Overall, the representation of the Coral Sea temperature structure and of the SEC jet temperature and transport structure in the three HighResMIP models matched the representation of the BRAN2020 ocean reanalysis and/or aligned with previous modelling-based and observation-based studies (Ganachaud et al., 2008, 2011, 2014; Kessler & Cravatte, 2013b). The HighResMIP models resolved the processes important in the formation of the NVJ, NCJ and SCJ, which includes the basin-scale wind stress and the blocking effect of Vanuatu and New Caledonia (i.e., Island rule, Kessler & Cravatte, 2013b).

There were some model-specific divergences in the representations of the SEC jet transports. For example, the transport maximum of the NVJ in FGOALS was too far north and the surface and subsurface maxima of the NCJ in CMCC were too strong. These divergences could have possibly arisen from differences in the local wind stress fields, or the mixing related to the vertical discretization scheme which are common sources of bias in models (e.g., Richter, 2015; Richter & Tokinaga,

2020). Furthermore, differences in model bathymetry clearly impacted the representation of the SCJ. There are few studies documenting the nature of the SCJ, probably due to the difficulty in observing its sub-surface structure amongst the considerable amount of eddy activity in the region (Cravatte et al., 2015), and this renders it difficult to make further comparisons with the literature.

The ENSO-related temperature and transport response of the SEC in the HighResMIP models also generally compared well with BRAN2020, observations (Kessler & Cravatte, 2013a; Wang et al., 2017) and modelling studies (Cravatte et al., 2021). The HighResMIP models generally captured the expected decrease in SST in the NVJ and NCJ region (i.e., the northern Coral Sea and around New Caledonia) following El Niños, and increases in SST following La Niñas (Cravatte et al., 2009; Wang et al., 2017). Additionally, in the HighResMIP models, NVJ and NCJ transport varied with ENSO, but SCJ transport did not. This aligned with the probabilistic modelling results of Cravatte et al. (2021) who found that the interannual variability in the transport of the NVJ and NCJ was largely deterministic, and related to ENSO, while the interannual variability in SCJ transport was largely chaotic. One caveat is that FGOALS only captured the correlations and monthly lags of the NVJ response. Interestingly, the Niño 3.4 index anomalies associated with El Niño and La Niña events in all three HighResMIP models were smaller than observed anomalies. This reflects the considerable uncertainty in the representation of ENSO by CMIP6 GCMs and in the projected changes to the ENSO forcing in response to climate change (Guilyardi, 2006; Lough et al., 2011; Cai et al., 2014; Risbey et al., 2014).

## 6.3 Projected 2050 changes

Looking to the future projections, substantial warming in the top 300 – 400 m of the water column was projected at the scale of the Coral Sea and in all currents, in all three HighResMIP models. The projected warming signals of a 4% to 7% increase in the SST of the SEC jets, relative to the historical means, were an order of magnitude greater than the historical variability in the SST associated with El Niño and La Niña conditions (ranging from 0.2% to 0.9%; Table 2, Table 3, Figure 3). Similar to recent studies of the global warming trend (Cheng et al., 2022), we found an acceleration of the warming trend in recent decades, with a doubling of the rate of SST warming from 0.09°C decade$^{-1}$ prior to 1985, to 0.21°C decade$^{-1}$ afterwards. The later faster rate is comparable to the SST trend observed between 1993 and 2021 (E.U. Copernicus Marine Service Information [CMEMS]).

Interestingly, a band of cooling lay below the warming signal. The movement of subsurface cooler, less salty water into the tropical Pacific has been observed and modelled (Jiang et al., 2024; Ju et al., 2022). The cooling originates from the eastern subtropical Pacific where the poleward migration of outcropping isopycnals results in the subduction of colder water masses. The cold anomalies then propagate to the tropics along the 25-26 kg m$^{-3}$ isopycnal surfaces of the subtropical gyre (Ju et al., 2022). The pattern of temperature change in the Coral Sea could be driven by the same mechanisms; however, additional regional analysis following Jiang et al. (2024) and Ju et al. (2022) would be required to confirm this.

The transport of the SEC jets was also projected to change before 2050, where the low latitude SEC jets, the NVJ and the NCJ, were projected to weaken, while the higher latitude SCJ was projected to strengthen. The magnitudes of the changes were equal to the changes in current strength associated with ENSO events. These changes were already established at 1.5°C global warming and remained stable at 2°C warming. As the globe is approaching 1.5°C warming, it would be informative to investigate if the projected changes in SEC jet transport relative to 1950 – 1979 are present in observations and reanalyses.

Trade wind changes (Sen Gupta et al., 2012), a poleward shift in the SEC bifurcation latitude (Zhai et al., 2014) and warming-driven surface acceleration (Peng et al., 2022) are among the factors linked to SEC transport variability and could form the focus of future research.

## 6.4 Biological impacts

Substantial temperature increases were projected for the Coral Sea and SEC jets. The Coral Sea houses expansive coral reef habitats, including the Great Barrier Reef (GBR), and elevated water temperatures have increased the frequency of coral bleaching (e.g., Harrison et al., 2019; Hughes et al., 2017). The projected warming will likely exacerbate heat stress and bleaching. All three HighResMIP models exceeded the thresholds of 1.5°C and 2°C global warming by mid 20[th] century and a majority (70 – 90%) of extant tropical coral reefs will very likely disappear even if global warming is constrained to 1.5°C

(Hoegh-Guldberg et al., 2018). Indeed, at 1.5°C and 2°C global warming, respective SST anomalies of 0.78°C ± 0.04°C SD and 1.12°C ± 0.06°C SD, as well as heating down to 300 to 400 m, were projected for the Coral Sea in the HighResMIP models. Furthermore, the increase in stratification associated with the surface warming across the Coral Sea could have additional detrimental impacts on ecosystems. More stratified waters can be less ventilated with reduced oxygen concentrations and nutrient fluxes, which can in turn impact marine productivity (Breitburg et al., 2018; Fu et al., 2016).

Beyond sea surface changes, we found that surface/subsurface temperature increases were generally accompanied by underlying bands of cooling. Furthermore, in two of the three HighResMIP models, the area of greatest warming in the NVJ and NCJ overlapped with a surface/subsurface strengthening of the current, resulting in increases or stability in the heat transported in the upper water column, and decreases in heat transport below. The NVJ and NCJ bifurcate in western boundary currents along the continental shelf which houses the outer reefs of the GBR. The jets are a vector of transport of Coral Sea

water into the GBR, as they intrude through reef passages (Benthuysen et al., 2016; Brinkman et al., 2002; Choukroun et al., 2010; Schiller et al., 2015), and may also influence the lagoonal circulation in some places (Gurdek-Bas et al., 2022). Their subsurface waters can be upwelled, sometimes providing cold water refugia from bleaching stress for the corals in the outer GBR (Riegl and Piller, 2003; Spring and Williams, 2023). The projected changes in heat transport with depth in the NVJ and NCJ could affect these upwelling dynamics. The upwelling of water on the outer shelf, and movement of Coral Sea water into

the GBR are fine-scale processes (10s of m to a few km) constrained by sharp bathymetry changes and reef topography, including inter-reef passages (Benthuysen et al., 2016; Brinkman et al., 2002; Choukroun et al., 2010; Schiller et al., 2015). Even the highest resolution participants of the HighResMIP experiment are too coarse to represent these fine-scale shelf break

processes. Furthermore, tides can be important drivers of upwelling and can move water through reef passages. CMIP6 models, including those in HighResMIP, do not include tides. Therefore, dynamic downscaling, inclusive of all relevant processes, would be needed to represent the interaction of the SEC jets with the shelf break. Coastal climate downscaling to understand how the projected changes will propagate into and impact the expansive ecosystems in the GBR will be the subject of future research.

## 6.5 Modification of western boundary currents

The projected changes in strength of the SEC jets could have flow-on effects for the western boundary currents at the Australian continental shelf. The main body of the NCJ and the northern branch of the NVJ form the Gulf of Papua Current (GPC) which flows northward through Torres Strait before turning eastward as it approaches Papua New Guinea (Kessler and Cravatte, 2013b; Ridgway et al., 2018). An offshoot of the NVJ also feeds directly into the GPC near the tip of Papua New Guinea (Ganachaud et al., 2014). Additionally, the southern branch of the NVJ and the SCJ form the East Australian Current (EAC) which flows southward down the east coast of Australia (Choukroun et al., 2010; Kessler and Cravatte, 2013b). It is important to understand future changes to the GPC and EAC because, as with other western boundary currents, they transport water meridionally, influencing weather patterns and modes (Hu et al., 2015, 2020) and affecting the temperature regime of coastal ecosystems (Bennett et al., 2015; Johnson et al., 2011). They also both feed into global ocean circulation features (Hu et al., 2015; Kessler and Cravatte, 2013b; van Sebille et al., 2014). The GPC, EAC, and EAC extension are projected to strengthen after 2050 (Hu et al., 2015; Oliver and Holbrook, 2014; Sen Gupta et al., 2012, 2021) in reaction to a weakening of equatorial trade winds and strengthening of south easterly trade winds after 2045 (Hu et al., 2015; Sen Gupta et al., 2012; Wang et al., 2014). However, our study suggests that changes in the SEC jets could drive changes decades sooner. More research is required to investigate the interplay between the potential changes in trade winds, SEC jets, and western boundary currents before 2050.

**Code/data availability**

CMIP6 HighResMIP data are available from the ESGF nodes (https://pcmdi.llnl.gov/CMIP6/). BRAN2020 data are available from the NCI data catalogue (https://geonetwork.nci.org.au/geonetwork/srv/eng/catalog.search#/metadata/f9372_7752_2015_3718).

## Author contributions

All authors conceptualised the study. CL secured funding and provided supervision. JS and CL developed the methodology. JS curated the study data and formally analysed and visualised the data. JS and CL prepared the original manuscript, and all authors contributed to the manuscript revision.

## Competing interests

The authors declare that they have no conflict of interest.

## Acknowledgments

The authors are grateful for the computing recourses available from the Australian National Computing Infrastructure (NCI). We would particularly like to thank Syazwan Mohamed Ridzwan from NCI for acquiring the CMIP6 HighResMIP data. We are thankful to Ajitha Cyriac and Chaojiao Sun for our fruitful discussions. JS was supported by a CSIRO Early Research Career Postdoctoral Fellowship.

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
