# Peer review of "The Historical Representation and Near Future (2050) Projections of the Coral Sea Current System in CMIP6 HighResMIP"

_EGUsphere, 2025_

## Author Response (AR1)

**egusphere-2025-171**

**Response to reviewers**

Dear editor,

Thank you for giving us the opportunity to submit a revised version of our manuscript titled: 'The Historical Representation and Near Future (2050) Projections of the Coral Sea Current System in CMIP6 HighResMIP' for publication in *Ocean Science*. We appreciate the time you and the reviewers have spent evaluating the manuscript, and we feel the work has benefited greatly from your efforts. Please find below our detailed responses to the reviewers comments. The reviewers comments are in blue, our responses are in black, and the line references in our responses refer to the revised manuscript in track changes, all markup. With track changes all markup, the line numbers are in ascending order but do sometimes jump between pages (known word issue).

Thank you for considering our work.

Best wishes,

Dr Jodie Schlaefer

jodie.schlaefer@csiro.au

**Reviewer 1:**

The paper uses data from some high resolution climate model runs to estimate the changes to expect in ocean currents and temperatures between now and 2025. The results may be useful for groups planning climate change mitigation studies.

Overall the writing was filled up with too much detail, much of it poorly expressed. What is lacking, or discussed much too briefly, is insight, especially what is learnt that was not known before. There is also little comparison with the results from lower-resolution models which would help in understanding the utility of the high-resolution model results.

Unfortunately I found the paper poorly written and often skipping important information. Also unfortunately I was unable to see any of the figures in the supplement as the files were in the close Microsoft *.docx format instead of an open format like *.pdf.

Detailed comments are given below. I recommend a major revision.

Thank you for taking the time to review the paper thoroughly. Your comments were really useful in guiding where the manuscript could be streamlined, and where text needed to be reworded to clarify our meaning. We have removed unnecessary detail and highlighted the insights gained from our work.

We did not compare the representation of the oceanography of the Coral Sea in low-resolution versus high-resolution models, because low resolution models do not realistically represent the SEC jets. The blocking effects of the large islands are a first-order feature of the southwest Pacific that control the jets separation, and the details of the bathymetry and passages control the split between the jets and ultimately control their latitudinal position (see uncertainties in the island rule calculation in Kessler and Cravatte (2013)). Literature about the SEC jets uses mainly ¼ or higher resolution ocean model (e.g., Cravatte et al. 2021). Fig S10 in SenGupta et al., 2021 clearly shows the limitation of low resolution in the definition of the coastlines in the complex western pacific area. The requirement to use high resolution models was fundamental to the study design and was explained in the introduction. Please see the expanded explanation in lines 131 to 138.

[Figure]

Figure S 10 Model representation of coastlines. Land-ocean masks over the maritime continent for three CMIP6 models (NESM3, MIROC-ES2L, FIO-ESM) in comparison to the coastline derived from the ETOPO2 dataset[i] (top left).

Response Figure 1. Figure S10 from Sen Gupta et al. (2021) which demonstrates how coastlines are poorly resolved in low-resolution global climate projections.

L 82.  Are there jets north of 10S?

We have added the jets that are present north of 10°S to Figure 1a following Ganachaud et al., 2014 (their Figure 2 is copied below): the South Equatorial Counter Current (SECC), another branch of the SEC north of the SECC, and the New Guinea Coastal Undercurrent (NGCU). We have also added the counter currents in the Coral Sea: the Coral Sea Counter Current (CSCC) and the South Pacific Subtropical Counter Current (STCC). Finally, we amended the path of the East Australian Current (EAC) to flow along the edge of the Great Barrier Reef, and over the Australian continental shelf.

We have modified the text to clarify that our paper focuses on the Coral Sea and the SEC jets reaching the Coral Sea. Please see lines 90 to 91.

[Figure]

Response Figure 2. Copy of Figure 2 from Ganachaud et al., (2014).

The quality of the all figures is poor.  They looks like jpeg copies, maybe screen copies, taken from a standard plotting package but really need to be a good quality vector graphics images.

In figure 1, the outline of the area used for the integrals later needs to be shown at large scale in figure (a).  The caption should not redefine NCJ etc.  Why are the differences in the heat content of the models and observations much larger than any of the variability?

Your concerns of figure quality were based on the blurred figures visible when the manuscript was viewed directly from the pre-print server. We assure you the figures are of high quality. Please see the revised text.

Within Figure 1a, the representation of the slices taken through the current jets for the integral analysis have been modified for clarity. Specifically, the slices are now jet-specific to reflect the jet-specific analysis, they have been widened and the colour has been changed to blue solid lines so they stand out more.

We have removed previously defined abbreviations from the figure caption as suggested.

The difference in heat content comes from different representations of the temperature distribution in the different models. Figure 3 c, g k highlights the warmer deep ocean layer in FGOALS, resulting in a higher ocean heat content than the other models in the top 1000 m of the Coral Sea. The magnitude of the modelled differences in temperature distribution are standard among different CMIP6 models (e.g., Khosravi et al., 2022; Wang et al., 2024). As shown in response Figure 3 below, model alignment in OHC improves when the calculation is made over a) 200 m, and b) 400 m, where there is less difference in the modelled temperature distributions.

In the manuscript, we have replaced the 1000 m OHC calculation with the 200 m calculation. This was done so we could better visualise the impact of the projected warming on the OHC, as the projected warming was greatest in the top 200 m over the studied period (see revised Figure 1b). We also acknowledge that the link between OHC and temperature distribution was not explicitly mentioned in section 4.1, and we have updated the manuscript to clarify this (see lines 684 to 688).

[Figure]

Response Figure 3. Annual average ocean heat content of the top a) 200 m, b) 400 m and c) 1000 m of the Coral Sea.

L 107. The southeast trades push the ocean westwards, they are not going to shift anything eastwards. Winds cool the ocean, so reduced winds will not cool it.

The west Pacific warm pool is normally defined as lying north of the Equator. Is this what you mean?

We acknowledge that we poorly described the relationship between the equatorial trade winds, the western Pacific warm pool and ENSO. The description of ENSO has been removed following the recommendation of reviewer 2.

L 152. The properties of BRAN2020 need to be described. Why was it used? Are there other datasets which could be used? If so how do they compared?

We have added a description of the BRAN 2020 ocean reanalysis and included a justification of our choice to use it. Please see lines 590 to 596.

L 164. It would be better if the model details were placed in a table. This should also use information on the near surface ocean layer thicknesses, the horizontal and vertical mixing scheme used, and the boundary conditions (slip/noslip) at horizontal boundaries.

The model details have been removed from the text and added to a table, along with information on the near surface ocean layer thickness which is most pertinent to this study. Please see lines 182 to 203.

L 178. Instead of writing out the long names of everything in full, it would be best if these were defined in a table with the text used for a short explanation of the difference between SSP-5-8.5 and SSP2-4.5 as everything else seems constant.

L 183. (37 models; figure 2). This seems to be some shorthand for "(27 models) and you must now concentrate on figure 2 because the next few lines are going to be about this figure.". This really needs a new paragraph or even a new section starting "Figure 2 ...".

L 188. So far the paper has been about SST. Does this line refer to SST?

This paragraph does not explain what 1850-1879 data was used, it does not need to tell the reader that is 30 years. It should mention that it is using the average over this period.

How were the averages calculated? What model fields were used.

Your comments highlighted the need to re-phrase part of section 2.2. We have rephrased the section to improve readability and to clarify the data used and how the averages were calculated. Please see lines 205 to 217.

L 189. Why could you not calculate the offset from the 1850-1879 average? What was it about the initial conditions that meant this could not be done?

The HighResMIP experimental design stipulated that the participating models would be run from 1950 to 2050. Therefore, data were only available from 1950 onwards. Please see line 207 and line 212.

L 192  Figure 2.  Quality problem as before.

Better colours are needed.  The dotted yellow colour is not clear.

The caption does not need "Shared Socioeconomic ..." etc.

A description of how the 'observations' was calculated would should be in the text.

Your concerns of figure quality were based on the blurred figures visible when the manuscript was viewed directly from the pre-print server. We assure you the figures are of high quality. Please see the revised text.

The caption has been shortened as suggested (lines 220 to 225), and a description of how the observations were calculated has been added to the text (lines 213 to 217).

L 201-225  This section needs rewriting.

The main point appears to be that, on the basis of the low resolution models, the 5-8.5 forcing is too strong but the reduced response of FGOALS is probably most realistic of the tree high resolution models.  However the way you get there is very confusing.

As this paper is about the ocean, why do you not also compare SST from the different models?

On the question of acronyms, you might like to find something shorter for FGOALS and HighResMIP.

Following discussions with the CMIP community, it became clear to us that it was very important to put the HighResMIP experiment into perspective with the larger CMIP6 ensemble. This is because HighResMIP was only introduced in CMIP6, it only included SSP5-8.5 and the historical period only starts in 1950

The main purpose of the comparison was to determine if the global warming signal in the HighResMIP models was within the range of the broader CMIP6 ensemble and could be trusted. Then, on establishing the global warming signal could be trusted, in later sections we move to looking at the representation of the regional oceanographic processes in the Coral Sea and the regional climate signal.

We agree that this message was lost within the large amount of detail provided in the text. The section has been greatly shortened for clarity. Please see lines 326 to 333.

We used the global-average near-surface air temperature in this section because it is widely used to quantify the global warming signal and was sufficient to build confidence in the global warming signals of the HighResMIP models. Adding a comparison of global SST would be complementary; however, we have opted to not add the additional analysis to this already substantial body of work. Comprehensive regional SST analyses are presented later in the paper.

We chose to retain the use of HighResMIP and FGOALS, which are the acronyms used by the CMIP community.

L 235 "energy and mass budgets". I suspect "heat and salt budgets" would be less misleading.

"Energy and mass budgets" has been revised to "heat and salt budgets" throughout the text.

L 248 Write down the equation in mathematical form and define the various terms after the equation.

As suggested, we have changed the order so the equation is stated before the terms are defined. Please see lines 350 to 437.

L 252 Shouldn't the integral be over latitude and longitude?

OHC is clumsy. Why not just 'H' or 'H' with some subscript.

It is the volume integral, so it is over latitude, longitude and depth. Please see line 435.

Choose to retain OHC as it is a commonly used acronym for Ocean Heat Content.

L 254 Why is this figure placed in the supplement - which I could not see.

Previously you have used scenario runs, presumably to distinguish them from the control runs. Here it is the control scenario and SSP5-8.5 scenario. Why not use 'run' - and as you only use one scenario, why give the extended name?

The figure explained part of the methodology, so it was placed in the supplement. The figures in the main text are focused on the results. We have replaced scenario with run as suggested. Please see lines 435 to 445.

 What does cell-wise mean?

Briefly summarise Iving et al (2021).

How did you calculate the coefficents to use in the CMCC correction?  Did these vary with position and depth?  Presumably the drift changed with time.  Why use a linear approximation and not just subtract the corresponding drift each month?

'Cell-wise' means for each model grid cell. We have changed it to 'grid cell wise'. The drift did vary with position and depth, and this was accounted for in the grid cell wise calculation.

Control and scenario runs are independent of each other, with their own evolving internal variability. As the variability is independent, the drift each month in the control run cannot directly be taken from the scenario run. We visualised the drift in heat and salt extensively before implementing the correction, and it was approximately linear through time. We used a well-established de-drifting procedure, common in the field of global climate model analysis.

We have added additional information on Irving et al. (2021) and on how we calculated the coefficients. Please see lines 439 to 442.

L 269  Why two figures, one in the supplement?

Figure 1a showed the broad location of the slice through all three current jets on the stylised current map. Figure S2a showed the location of the jet-specific slices on the 1000 m depth averaged current field from BRAN2020.

We have updated Figure 1a to include the jet-specific slices. We feel it is informative to retain the visualisation of the 1000 m depth averaged current fields from each model in the supplement.

L 270  What is a focal current?

Focal currents referred to the currents we focused on in the study.

We have changed:  "The slices were restricted to the top 1000 m of the water column where the main bodies of the focal currents flow."

to

"The slices were restricted to the top 1000 m of the water column, accounting for the majority of the transport of the surface -intensified jets"

See lines 451 to 452.

Removed throughout the text as suggested.

L 272 I expect the jets to be eddying, It would be nice to see a typical distribution of SST in at least one of the models, which can be compared with the corresponding control SSH field.

In calculating heat transport, the best estimate is the integral over time of velocity time temperature. If you are working with monthly average values this misses a lot of the signal but the archive datasets often include a heat flux field. Do any of the models have such a field?

Assuming you do not have access to such fields, why use a mask so that only westward flows are used. What is gained over just using a latitude mask for each current jet?

CMIP ocean fields are stored as monthly-means, smoothing eddy activity in the datasets. The 'Ocean Heat x transport' variable, hfx, was only available for the CMCC model. We chose a method that we could apply to all models.

We focus on the predominant westward portion of the flow as it transports the climate signal through the Coral Sea to the Australian continental shelf break and into Western Boundary Current. We have added these details to the text, please see lines 458 to 459.

Our ultimate goal is to run a regional climate downscaling to look at the future of the western boundary currents and their variability (subject of future paper). Here we assess the monthly heat transport that will be used as boundary condition for such a modelling experiment.

Revised as suggested.

L 277 A mathematical equation properly written out would be better.

We have added a formula added as suggested, please see lines 460 to 464.

L 282 "Specifically ..." -> "The data was decomposed into seasonal and interannual components and a long term trend"

Revised as suggested (line 498).

 Equation for seasonal decomposition - here or in an appendix.

Only 1% of readers will know what the STL function is. Again give the equation .

Similarly with LOWESS.

We have provided details on the functions used so others can easily reproduce the methodology, which is a common practice in scientific writing.

We have provided the equation for the temporal decomposition and feel including the separate equations for the individual functions will add a level of detail that will hamper readability and understanding.

L 289  The logic of the text indicates that epsilon should be zero.

We have revised the text to clarify where the residual came from. Please see lines 498 to 504.

L 290  "The same ...".  Unneeded.

Removed as suggested.

L 293  "was not explored" - why include this.

Removed as suggested.

L 294  Here you say the Nino 3.4 index was calculated for BRAND ...

Thank you for catching this error, we have removed it.

L 299  Here you say it was not necessary.  Why is it not necessary if it is a reanalysis product?

It was not necessary as BRAN2020 is a reanalysis product, so timing of El Nino's/La Nina's in BRAN2020 tracks with the observed Nino 3.4.

L 306  Again I would like to see some equations, either here or in an Appendix.

Cross-correlation is a standard statistical procedure. We believe adding the equation will add a level of detail that will hamper readability. We have added additional details to the text to improve the reproducibility of the method, including stating the python function and version

used, please see lines 512 to 518439. The results of the cross correlation are included in the appendix.

 In the main body of the text the use of mathematical symbols, except when they are being defined, is bad. What does it sound like when you try to read this section aloud?

The mathematical symbols have been removed from the text, except where they are being defined, as suggested.

 What does 'isolated trend profiles' mean.

We have changed this to 'trend components' (line 521).

 J/s. Not the place to say that you are using SI units.

We retain the use of units here for previously undefined variables to remain consistent with the rest of the text, and because it helps with reader understanding.

 If you have defined variables correctly earlier in the manuscript you do not need to repeat here.

Variables that were defined earlier have been removed as suggested (line 524).

 If you need to define a variable, do it properly, not in this way.

Revised as suggested, please see lines 524 to 528.

 Within the main text expand variables such as $U_{trend}$ in words.

 Ditto. In the following I will not repeat the comment.

Revised throughout the text as suggested.

 Only 1% of readers will know what the gsw toolbox is.

We have provided details on the function used so others can easily reproduce the methodology, which is a common practice in scientific writing.

L 337-338  Paragraph need rewriting.  "Therefore" is out of place.

Therefore has been removed as suggested.

L 341  What is meant by 'scale'

We have clarified that in this section we are doing a regional assessment of the historical representation of the Coral Sea in the HighResMIP models (line 588 to 589).

We have removed 'scale'.

L 342  What is it about Fig 1b which supports this statement?  For a start the differences seem to be much bigger than any drift or the size of an short term fluctuations.

We have revised the OHC analysis to focus on the top 200 m of the water column. Please see the revised Figure 1, and revised text at lines 684 to 688.

L 344  Why is this statement about the jump here?  Why is it relevant?

We have removed this statement.

L 346 'well represented'.  Justify the use of 'well'.  All I see is the normal exponential decay with depth plus some kinks.  Are there similar plots from low resolution runs of similar models?

We state 'well represented' in the topic sentence and provide justification in the paragraph that follows. In particular, we discuss the representation of the latitudinal temperature gradient, vertical structure and mixed layer.

We did not compare the representation of the oceanography of the Coral Sea in low-resolution versus high-resolution models, because low resolution models do not realistically represent the SEC jets. We have provided further details in our response to your overall comment. We have expanded the text in the introduction where we explained the requirement to use high resolution models. Please see lines 131 to 138.

L 354  The models are now not so good.

This is a description of small differences within a generally good representation.

L 355 'shared the characteristics with'. What characteristics are shared? How can the statement be checked?

Revised as suggested (lines 607).

L 357 Is this profile in Ganachaud.

"derived from ... in", not needed

Revised as suggested, please see line 682.

L 358 "Overall ... " - not proven.

We have changed 'overall' to 'in summary', please see line 687.

L 366 What does 'historical' mean here.

We have changed 'historical mean' to 'climatological', and the date range of the climatology is provided at the beginning of the figure caption. Please see lines 705 to 709.

L 368 How do the crosses within a circle indicate direction?

We have removed the circled crosses from all figures, and are using 'positive eastward' to indicate the direction following a comment from reviewer 2. Please see line 707.

L 379 This way of representing the change due to El Nino/La Nina does not really work. Something better is needed.

We have revised the table as suggest. We have split the historical assessment results (Table 2) and future change results (Table 3) into two separate tables. In Table 2, we have modified it, so the sign of change with El Nino's is show, and the caption states the change with La Nina is the opposite sign. Please find Table 2 at line 715 and Table 3 at line 1040.

L 389 Use no space or non-breaking space between dimensions.

Revised as suggested.

L 392 'aligned with' - too vague for a scientific paper.

transport/speed - choose one.

We have changed 'aligned with' to 'were similar to'. We have change 'transport/speed' to 'transport and speed'. Please see line 828.

L 392 ' diverged' - usually 'diverged from' something.

Sentence need tidying up.

Revised as suggested. Please see line 828.

L 402 It sounds as if these figures should be part of the main text.

The additional heat transport calculations are not a major finding of the paper, so we will retain the placement of this figure in the supporting information.

L 416 - 452. I cannot comment on details as the I do not have access to the figures. However I am do not understand the point of these detailed descriptions: What is learnt that is important? What will be important when the paper considers changes over the next 25 years?

In section 4.2 and 4.3, the jet characteristics in BRAN2020 are briefly described, the HighResMIP model jets are benchmarked against BRAN2020, and the jet characteristics are contrasted with the literature. This is essential to establishing that the HighResMIP models are able to represent the complex hydrodynamics of the Coral Sea. It was important to build this confidence before we looked at what the models project for the future. We have clarified this in lines 588 to 589.

Details have been removed where possible; however, we feel it is important to retain most of this text. Please see the revised sections.

L 455 As written this sentence may not be referring to BRAN. Again what will be important later?

Revised as suggested (line 949). Please see previous comments.

 Having previously indicated that for global surface temperatures FGOALS is best - why average the three models here?

We averaged results across the HighResMIP models in section 5, despite the difference in global warming projections because 1) we established that all the HighResMIP models reasonably represented the oceanography of the Coral Sea in section 4 and 2) it was informative in the broader context of CMIP considering the global warming projections of the three HighResMIP models represented a range of projections within the CMIP6 ensemble.

L 489 What does 'simulated to be carried' mean here.  Why is the first sentence needed?

We have greatly revised this section, please see lines 1030 to 1038.

L 490 "over the 100-year period from mid 20th century to mid 21st century"  !!

Changed to 'from 1950 to 2050'. Please see line 1031.

L 491 Figure 1 does not show significant changes in heat content.

We have revised the figure to include the OHC calculated from the top 200 m of the water column. This better shows the projected increase in OHC.

L 493 This is one of the main sections of the paper but it concentrates too much on details.  Given the model results and the climatology, what do the authors think are the key changes?

We have removed excessive detail from section 5.1 to focus on the key changes as suggested. Please see lines 1030 to 1144.

L 514 "There are marked changes ... under 1.5 and 2 C ...".  Looking at figure 5b there looks as if there is an almost linear relationship between the global temperature anomaly and the change in near surface ocean temperatures.  Would the agreement be even better if a comparison was made with the global SST anomaly or the tropical SST anomaly?

The SST warms slower than the Air temperature over land (see 4[th] figure https://www.carbonbrief.org/state-of-the-climate-2023-smashes-records-for-surface-temperature-and-ocean-heat/) and both warming rates are rather linear.

We used the global average near surface air temperature in the comparison of global warming projections as it is the variable most commonly used to quantify global warming (in particular the 1.5°C and 2°C thresholds). In figure 2, the main purpose of the comparison was to

determine if the global warming signal in the HighResMIP models was within the range of the broader CMIP6 ensemble and could be trusted. The use of the global average near surface air temperature was sufficient to build confidence in the global warming signals of the HighResMIP models.

We then use the well-established 1.5°C and 2°C thresholds to investigate the projected future changes in the Coral Sea.

We have added a comparison of the Coral Sea SST anomalies from the HighResMIP models and the observed anomaly, please see the discussion at lines 1345 to 1349.

L 559  What does 'projected' mean here and in line 661.  Is the first sentence needed?

The first sentence is a topic sentence that sets the focus for rest of the paragraph. Projected is used here as CMIP6 are projections, and section 5 focuses on what the HighResMIP models project for the future period.

L 562  How does "projected near future decreases" differ from "predicted drop"

Projected is used here as CMIP6 are projections, and section 5 focuses on what the HighResMIP models project for the future period.

L 566  Why are the words 'ensemble mean' needed here.

Removed as suggested. Please see line 1201.

L 570  Intensification of what?  What am I supposed to see in Fig 7 which is not in Fig 6?

This sentence has been revised, please see lines 1203 to 1204. Figure 6 shows the ensemble mean change, and Figure 7 shows individual model results of models that projected a near surface transport intensification of the NVJ and NCJ.

L 571  The manuscript needs a better description of Figure 7.  What prompted the split into an upper and lower layer and how are these layers connected with 'a minor shallowing'.

We have added a description of figure 7, please see lines 1208 to 1212.

  Where are the major conclusions from section 5?

We have heavily revised section 5 to highlight the major conclusions. They are also discussed in section 6.3.

L 599  After finding the previous sections of the paper hard work, I found this final section much better.  I would have liked to see some comparison with the results from lower resolution versions of the same models to see how much the results are dependent on improved resolution.

   The biological section contains a few too many motherhood statements but one result of the present work which I find unexpected is the increase in cold water at depth.  I would welcome more on why it occurs.  Globally we expect surface heating to penetrate more with time, as is found in the results - so more cold water close to the surface in the Coral Sea is an anomaly which is worth is bit more understanding.

We did not compare the representation of the oceanography of the Coral Sea in low-resolution versus high-resolution models, because low resolution models do not realistically represent the SEC jets. We have provided further details in our response to your overall comment. We have expanded the text in the introduction where we explained the requirement to use high resolution models. Please see lines 131 to 138.

We have reduced the biological impacts section. Please see the revised section 6.4.

We have added more information on the subsurface cooling following comments from reviewer 2. Please see lines 1350 to 1356.

**Reviewer 2:**

This work analyses the future of the South Equatorial Current and temperatures based on CMIP6 2050 projections. The authors select three climate model with a high resolution ocean and compare them with the BRAN reanalysis and published estimates of transports and temperatures, and their variations, before documenting their projections. I found the paper well-structured, and easy to follow. I recommend publication with minor revision.

Thank you for taking the time to provide a thoughtful and constructive review. The text has benefited greatly from your comments. Please find below replies to your specific comments.

Minor comments:

L83: SPG only 3 times used, would not introduce this acronym

We agree and have the removed the acronym as suggested. Please see lines 52 to 61.

**L105-110. not necessary – just provide a ref to ENSO.**

We have removed the text as suggested. Please see line 115.

**L111: specify what is "after" (3 months ? 3 weeks...?)**

Thank you for pointing this out, we have added that Kessler and Cravatte found a lag time of a few months. Please see line 116.

**L196: a transparent line... is not visible! Please correct, eg "grey".**

We have amended the wording to 'light and dark lines'. Please see line 222.

**L207-208: I am not sure I understand: does it make sense to compare HR models forced with SPP5-8.5 with the SPP2-4.5 models ? I would only compare HR models forced with SPP2-4.5, if needed.**

Following discussions with the CMIP community, it became clear to us that it was very important to put the HighResMIP experiment into perspective with the larger CMIP6 ensemble. This is because HighResMIP was only introduced in CMIP6, and it only included SSP5-8.5.

The main purpose of the comparison was to determine if the global warming signal in the HighResMIP models was within the range of the broader CMIP6 ensemble and could be trusted. Then, on establishing the global warming signal could be trusted, in later sections we move to looking at the representation of the regional oceanographic processes in the Coral Sea and the regional climate signal. The text in this section is now greatly reduced to clarify the purpose of the comparison.

We have modified the text describing the results to focus on the comparison between the HighResMIP projections and SSP2-4.5. We retain the SSP5-8.5 ensemble in the Figure to provide a broader context with CMIP6. Please see lines 326 to 333.

**L285: "leaving the interannual variability around the trend" what does «around the trend» mean? would «and the trend» be suitable?**

Revised as suggested (line 501).

**L289 is (equation): I would recommend tu use capital letters for T and S, as t is generally time, and s various, pdf, frequency**

We have replaced the lowercase t and s with capitals as suggested.

L306: how well do these models simulate El Nino et La Nina ? (Reference?)

There is limited literature on how well the selected HighResMIP models simulate ENSO. Our findings in regard to this are explored in results section 4.3 and discussed in section 6.2.

L319: I cannot follow - please specify, or provide explanation, eg we first replaced t_trend with the observed climatological trend in equation X, to evaluate...

We have revised the text as suggested. Please see lines 524 to 528.

L323-326: It is confusing to me. It seems that you identify the max with depth, but as you then use «peak», it is not clear wether it is a peak in depth or time; please uniformize, I would suggest using maximum instead of peak then.

We have uniformized the language to maximum. Please see lines 578 to 580.

L354: "Cool with depth" mislead me, as «cooling» is usually used as an evolution with time. How about using «temperature decrease with depth» to avoid this

Revised as suggested, please see line 606.

L369: not clear; just state «positive eastward»

Revised as suggested, and we have removed the circles crosses from all figures. Please see line 707.

Fig S8: legend: time-varYing , add «y»

Thanks, we've fixed the typo as suggested.

L406-407: If "observed" refer to the Ganachaud et al or Kessler and Cravatte ones, make it explicit, eg, "the BRAN 2020 matched the observed mean structure of the NVJ of Ganachaud et al; K&C "; specify -Fig 13g for Kessler and Cravatte: I suggest to add: something like «with the 0.1cm/s isotach close to 250m, compare Fig 4a to their their Fig13g»

Revised as suggested, please see lines 834 to 836.

Revised as suggested, please see line 870.

L417: replace the period with ", in comparison with that of the NVJ"

Revised as suggested, please see line 882.

L420-421: I see only one maximum between 200 m and 350m in their fig 13h

Thank you for catching this, we have revised the text to reflect this. Please see lines 884 to 886.

Revised as suggested, please see line 888.

l425-426: I would word this in an opposite way, eg, «the NCJ transport of 12 Sv in CESM and FGOALS underestimated by 25% the BRAN 2020 transport of 15Sv». (since the latter is your reference.)

Revised as suggested, please see lines 889 to 890.

427: Notably,... replace with «In CESM, the overestimated subsurface transport balances the underestimated deep transport»

Revised as suggested, please see lines 890 to 891.

l439-440: "What do you call «observations»? the Ganachaud/Kessler/Cravatte ? If comparing with Kessler and Cravatte 2013 Fig 13i, those are geostrophic currents, therefore, BRAN may carry Ekman components that are not in Kessler and Cravatte near the surface.

We have modified the text to be more explicit when comparing the model results to geostrophic velocities, and have added that the modelled currents would carry Ekman components that are not included in geostrophic currents. Please see lines 900 to 906.

We have also added extra velocity contours to Figure 4 to make it easier to compare the SEC jet representation with the literature.

L450: a comment on why this different finding would be welcome (different model/forcing, etc...)

We have added more detail on why the representation of the SCJ may differ so much between models, please see lines 910 to 944. We have also added a figure to the supporting information (Figure S10) and modified Figure 4 to better show the different bathymetries of the different models.

L456-459: "was cooling (warming) I recommend to ban this wording that constraints one to read the same sentence 3 times with risks of confusion. Just write anoter sentence for La Nina, eg, for La Niña, the effect is reversed. Also: «cooling «down to 1000m»

This description has been removed following comments from reviewer 1.

L460: «small» is quite subjective (even smaller in °Kelvin): is it significant ? (correlation >0.4) ? is it important ? If it is neither, I suggest skip the paragraph and just state there are little, insignificant variations with ENSO.

We have reduced this text and clarified that the correlation was significant, but the magnitude of the change may not be important (small relative to the mean), please see lines 953 to 956.

Added as suggested, please see line 950.

L474-476: confusing sentence:  I suggest to find a better wording.

We have reworded the sentence as suggested. Please see lines 959 to 961.

l526: the vertical profiles Fig S13a look like a depth-integral. Is it really a velocity anomaly? All velocity anomalies are westward in the upper 200m on Fig 13a in CESM too, but not in d, g, j.

The top row has climatologies, and the second to fourth row have anomalies. We have reworded the caption to clarify this, particularly replacing the variables with word descriptions in the text, please see the revised supporting information.

l532-534: this belongs to the discussion

We have moved this to the discussion as suggested, please see lines 1342 to 1345.

l536: I would be intersested to know if such magnitude is important an why

We converted the rate of stratification to a percentage so it could be contrasted with the findings of Li et al. (2020) to give a better indication of its importance. Please see lines 1140 to 1144.

l544: I do not understand the circle cross in relation with the graphic: positive eastward ? westward ?

We have removed the circled crosses from all figures, and now use: 'For transport, positive (brown) is a decrease westward and negative (blue) is an increase westward.' in the figure captions to explain the change with direction. Please see lines 1187 to 1188.

l550: The profiles of S15b look really different from those of S16e (and others). Is that all due to the fact that those are 1.5° or 2° whereas, the S15b are 2050 ?

The profiles in S15b are climatologies and the profiles in S16e are anomalies from the climatology. We have reworded the figure captions to clarify.

l582: need to label dash-dot/plain lines on i and j; define «positive eastward» if so.

The figure label and caption have been revised as suggested.

l596-597: not clear: please re-word

We have reworded for clarity, please see lines 1196 to 1197.

l599: Could you discuss sensitivity to the choice of definition of the transports lines min and max latitudes, Fig S2a. Actually this definition is so fundamental it should be on Fig. 1 too

We did not explore the sensitivity of the minimum and maximum latitudes of the current slices. The minimum and maximum latitudes of the jets were selected from Hovmöller diagrams of the zonal current speeds on the 162°E meridian, from which the jets were clearly visible. These details have been added to the methods. Please see lines 453 to 454.

L619-631: This all looks like a disperate justification of using SPP5-8.5 that is now considered unlikely because it would assume irrealistic political decisions and technological backing, where you could have used a more realistic scenario. I suggest to shorten the discussion, and just state that your projections are for 1.5° and 2°, regardless of the scenario.

We have greatly reduced this section to focus on the comparison with the SSP4-2.5 ensemble. Please see lines 1271 to 1282.

L650: More appropriate is Cravatte et al 2015, p. 269: «The coherent SCJ as seen in the long-term mean near-surface field (Fig. 4b) is most often masked by eddies aliasing synoptic surveys. « https://doi.org/10.1016/j.jmarsys.2015.03.004

Updated as suggested. Please see line 1326.

L656-657: Before you make this conclusion, you may want to check the meridional extent of the warm pool, as El Nino provides cooler SST near New Caledonia. I have this oldish reference, but there are certainly newer ones.: https://doi.org/10.1007/s00382-009-0526-7

We found complementary results, and have modified the text for clarity. Please see lines 1330 to 1333.

L663-664: you can't state this without further justification. What leads you to make this assumption for one model, and not the other ones?

L665-666: why?

Further investigation would be needed to determine why FGOALS only captured the ENSO response of the NVJ, so we have removed these statements. Please see line 1336.

L680: wind stress curl positive? Negative? anomaly; location ?

The text has been revised to clarify the mechanism of subsurface cooling, please see lines 1350 to 1356.

L722: As you write below, this should be substanciated, ie: the circulation in lagoons can also be independent of the large-scale currents, but depend mostly on local winds and tides. I suggest to just state that the influence needs to be studied to avoid being considered as «alarmist» versus «alarming».

We have revised this section, removing the speculative assertions. The text now just indicates that further research is needed to determine the influence of the projected changes on upwelling dynamics and lagoonal circulation. Please see lines 1412 to 1418.

**References**

Khosravi, N., Wang, Q., Koldunov, N., Hinrichs, C., Semmler, T., Danilov, S., and Jung, T.: The Arctic Ocean in CMIP6 Models: Biases and Projected Changes in Temperature and Salinity, Earth's Future, 10, e2021EF002282, https://doi.org/10.1029/2021EF002282, 2022.

Li, G., Cheng, L., Zhu, J., Trenberth, K. E., Mann, M. E., and Abraham, J. P.: Increasing ocean stratification over the past half-century, Nat. Clim. Change, 10, 1116–1123, https://doi.org/10.1038/s41558-020-00918-2, 2020.

Sen Gupta, A., Stellema, A., Pontes, G. M., Taschetto, A. S., Vergés, A., and Rossi, V.: Future changes to the upper ocean Western Boundary Currents across two generations of climate models, Sci. Rep.-UK, 11, 9538, https://doi.org/10.1038/s41598-021-88934-w, 2021.

Wang, Z., Brickman ,David, Greenan ,Blair, Christian ,James, DeTracey ,Brendan, and and Gilbert, D.: Assessment of Ocean Temperature Trends for the Scotian Shelf and Gulf of Maine Using 22 CMIP6 Earth System Models, Atmosphere-Ocean, 62, 24–34, https://doi.org/10.1080/07055900.2023.2264832, 2024.

---

## Referee Report (RR1)

Comments on revision 1 of "The Historical Representation and Near Future (2050) Projections of the Coral Sea Current System in CMIP6 HighResMPI"

The revised paper is much improved.  I am sorry that there is no explanation of the cooling at 400m but that is a minor point.

Some of the writing is still clumsy, verbose etc.  I have listed some points below but this side of the paper is best handled at the copy-editor stage.

Minor comments:

Line 34.  utilised -> used.
Line 35.  'captured' -> and found that these successfully represented...
Line 37.  'warming signal'.  This construct is used a lot later in the paper but always seems clumsy.  What you really mean is 'the maximum depth affected by the surface warming'.  Maybe there is a better way of saying this.
Line 39.   'with future global warming'.  Not needed.

Line 66.  Figure 1.  As this is a key figure readers skimming through the paper will find it useful to have the full names of NVJ, NCJ and SCJ, as with the other currents.

Line 77.  'in the context of' - bureaucrat!
Line 90.  'El Nino is no longer'.  El Nino has not changed, its people's belief that has changed.
Line 96-98.  The blocking effects of the islands is not a mesoscale process, although the scale of blocking and the mesoscale processes may be similar.

Line 112'  'The global degrees warming'  Clumsy.
          "We campare ... to ... to ..".  Usually one compares one
          thing 'with' another.
Line 113   'contextualise' - bureaucrat!

Line 135.  'The degrees warming'  !
Line 136.   ... used to quantify global warming - not global warming anomalies.
Line 144   'averaged'  Averaged over time?
           'visualisation'  -> plotted.
Line 156.  'degrees of warming'!!  Increase in temperature?

Line 178.  'Drift in the heat budget'.  In science a budget usually refers to an equation with inputs and outputs.  You could say 'The net effect of changes in the heat budget was determined...

Line 181.  This rewriting of an equation in words is clumsy.  The reader should know what an integral sign looks like.
Line 185.  'stable through time'.  Misuse of the word stable - none of

the models are expected to give infinite temperatures.  What you are
really comparing is the apparent long term change with the noise.  In
two cases the ratio is small and could be zero.  In the other it is
obviously non-zero.

Line 194.  'Notably'.  Not needed.

Line 198.  'accounting for' clumsy.

Line 208.  Multiplication signs are not usually used in equations unless
indicating a vector cross product.
Line 209.  Again the equation does not need specifying words, only the
variables need to be defined.

Line 215-230.  The python references should be replaced by references to
the original papers.  For SDL I think it is Cleveden 1990, for LOWEST,
Cleveden 1979.  The EGU journals request authors to include a link to
any computer (python) code used in the analysis.

Line 300.  'as it passed' - not needed.
           'corresponding' - not needed.

Line 387.  'projected' - clumsy.  Do you mean 'predicted'?

Line 566.  It is not correct to say that the jets 'force' the boundary
currents - as might be the case if the inertial terms in the momentum
equations were larger than the pressure gradient or coriolis terms.
Both the jets and the boundary currents are responses to the large scale
pressure gradients modified by the local limitations in flow due to
topography.

---

## Author Response (AR2)

**Response to reviewers, round 2**

Dear Professor Heywood,

Thank you for giving us the opportunity to make minor revisions to our manuscript titled: 'The Historical Representation and Near Future (2050) Projections of the Coral Sea Current System in CMIP6 HighResMIP' for publication in *Ocean Science*.

We have carefully gone through the paper and revised the wording where necessary. We appreciate the time you and the reviewers have spent evaluating the manuscript, and we feel the work has benefited greatly from your efforts. Please find below our detailed responses to the reviewers comments. The reviewers comments are in blue, our responses are in black.

Thank you for considering our work.

Best wishes,

Dr Jodie Schlaefer

jodie.schlaefer@csiro.au

**Reviewer 1:**

Comments on revision 1 of "The Historical Representation and Near Future (2050)

Projections of the Coral Sea Current System in CMIP6 HighResMPI"
The revised paper is much improved. I am sorry that there is no explanation of the

cooling at 400m but that is a minor point. Some of the writing is still clumsy, verbose

etc. I have listed some points below but this side of the paper is best handled at the

copyeditor

stage.

Thank you for taking the time to re-read the manuscript, and for highlighting where the writing needed clarification.

An explanation of the cooling at 400 m was provided in the discussion, section 6.3.

Line 34. utilised -> used.
Revised as suggested (line 34).

Line 35. 'captured' -> and found that these successfully represented...
Revised as suggested (line 35).

Line 37. 'warming signal'. This construct is used a lot later in the paper but always

seems clumsy. What you really mean is 'the maximum depth affected by the surface

warming'. Maybe there is a better way of saying this.
Revised as suggested (line 38).

Line 39. 'with future global warming'. Not needed.
Removed as suggested (line 39).

Line 66. Figure 1. As this is a key figure readers skimming through the paper will find it

useful to have the full names of NVJ, NCJ and SCJ, as with the other currents.
Revised as suggested.

Line 77. 'in the context of' - bureaucrat!
Revised as suggested.

Line 90. 'El Nino is no longer'. El Nino has not changed, its people's belief that has changed.

Revised as suggested.

Line 96-98. The blocking effects of the islands is not a mesoscale process, although the scale of blocking and the mesoscale processes may be similar.

Revised as suggested.

Line 112' 'The global degrees warming' Clumsy.

"We campare ... to ... to ..". Usually one compares one thing 'with' another.

Revised as suggested.

Line 113 'contextualise' - bureaucrat!

Revised as suggested.

Line 135. 'The degrees warming' !

Revised as suggested.

Line 136. ... used to quantify global warming - not global warming anomalies.

Revised as suggested.

Line 144 'averaged' Averaged over time?

'visualisation' -> plotted.

Revised as suggested.

Line 156. 'degrees of warming'!! Increase in temperature?

Revised as suggested.

Line 178. 'Drift in the heat budget'. In science a budget usually refers to an equation with inputs and outputs. You could say 'The net effect of changes in the heat budget was determined...

Revised as suggested.

Line 181. This rewriting of an equation in words is clumsy. The reader should know what an integral sign looks like.

Revised as suggested.

Line 185. 'stable through time'. Misuse of the word stable - none of the models are expected to give infinite temperatures. What you are really comparing is the apparent long term change with the noise. In two cases the ratio is small and could be zero. In the other it is obviously non-zero.

Revised as suggested.

Line 194. 'Notably'. Not needed.

Revised as suggested.

Line 198. 'accounting for' clumsy.

Revised as suggested.

Line 208. Multiplication signs are not usually used in equations unless indicating a vector cross product.

Revised as suggested

Line 209. Again the equation does not need specifying words, only the variables need to be defined.

Revised as suggested.

 The python references should be replaced by references to the original papers. For SDL I think it is Cleveden 1990, for LOWEST, Cleveden 1979. The EGU journals request authors to include a link to any computer (python) code used in the analysis.

We have added the paper references as suggested.

We have provided details of the functions we used, as well the parameterisations we used.

 'as it passed' - not needed.

'corresponding' - not needed.

Revised as suggested.

 'projected' - clumsy. Do you mean 'predicted'?

Projected is used here as CMIP6 are projections, and section 5 focuses on what the HighResMIP models project for the future period.

 It is not correct to say that the jets 'force' the boundary currents - as might be the case if the inertial terms in the momentum equations were larger than the pressure gradient or coriolis terms. Both the jets and the boundary currents are responses to the large scale pressure gradients modified by the local limitations in flow due to topography.

Revised as suggested.